# E-cadherin binds to desmoglein to facilitate desmosome assembly

Omer Shafraz[1†], Matthias Rübsam[2†], Sara N Stahley[3], Amber L Caldara[3], Andrew P Kowalczyk[3], Carien M Niessen[2], Sanjeevi Sivasankar[1]*

[1]Department of Physics and Astronomy, Iowa State University, Ames, United States; [2]Department of Dermatology, Cologne Excellence Cluster on Cellular Stress Responses in Aging-associated Diseases, Center for Molecular Medicine Cologne, University of Cologne, Cologne, Germany; [3]Department of Cell Biology, Emory University School of Medicine, Atlanta, United States

**Abstract** Desmosomes are adhesive junctions composed of two desmosomal cadherins: desmocollin (Dsc) and desmoglein (Dsg). Previous studies demonstrate that E-cadherin (Ecad), an adhesive protein that interacts in both *trans* (between opposing cells) and *cis* (on the same cell surface) conformations, facilitates desmosome assembly via an unknown mechanism. Here we use structure-function analysis to resolve the mechanistic roles of Ecad in desmosome formation. Using AFM force measurements, we demonstrate that Ecad interacts with isoform 2 of Dsg via a conserved Leu-175 on the Ecad *cis* binding interface. Super-resolution imaging reveals that Ecad is enriched in nascent desmosomes, supporting a role for Ecad in early desmosome assembly. Finally, confocal imaging demonstrates that desmosome assembly is initiated at sites of Ecad mediated adhesion, and that Ecad-L175 is required for efficient Dsg2 and desmoplakin recruitment to intercellular contacts. We propose that Ecad *trans* interactions at nascent cell-cell contacts initiate the recruitment of Dsg through direct *cis* interactions with Ecad which facilitates desmosome assembly.

DOI: https://doi.org/10.7554/eLife.37629.001

*For correspondence:
sivasank@iastate.edu

†These authors contributed equally to this work

Competing interests: The authors declare that no competing interests exist.

## Introduction

The formation, organization and maintenance of complex tissue structures are mediated by the cadherin superfamily of cell-cell adhesion proteins, a large protein group composed of four major subfamilies: classical cadherins, desmosomal cadherins, protocadherins and atypical cadherins (*Brasch et al., 2012*). Of these proteins, desmosomal cadherins and classical cadherins are essential for the maintenance of tissue integrity (*Rübsam et al., 2017a*). While classical cadherins mediate cell-cell adhesion in all soft tissue and play critical roles in tissue morphogenesis, desmosomal cadherins mediate robust cell-cell adhesion in tissues like the epidermis and heart that are exposed to significant levels of mechanical stress (*Dusek et al., 2007*). There are two types of desmosomal cadherins: desmoglein (Dsg) and desmocollin (Dsc), which are organized into four Dsg isoforms (Dsg1-4) and three Dsc isoforms (Dsc1-3) (*Getsios et al., 2004*). Of these isoforms, Dsg2 and Dsc2 are widely expressed in the epithelia and are also found in non-epithelial cells such as in the myocardium of the heart and lymph node follicles (*Getsios et al., 2004*). Loss of Dsg2 is embryonically lethal (*Eshkind et al., 2002*) while mutations in Dsg2 (*Awad et al., 2006*) and Dsc2 (*Syrris et al., 2006*) cause arrhythmogenic right ventricular cardiomyopathy (ARVC), a hereditary heart disease. In contrast, isoforms 1 and 3 are restricted to complex epithelial tissues (*Green and Simpson, 2007*) and their loss of function leads to epidermal fragility, such as in the autoimmune blistering disease pemphigus (*Amagai and Stanley, 2012*).

Dsc and Dsg associate with anchoring and signaling proteins to form robust intercellular junctions called desmosomes. Several studies have shown that the classical cadherin, E-cadherin (Ecad), promotes desmosome assembly. Immuno-electron micrographs demonstrate that Ecad localizes to the intercellular region of the bovine tongue epithelial desmosomes (*Jones, 1988*). Blocking Ecad adhesion with antibodies delay desmosome formation in MDCK cells (*Gumbiner et al., 1988*) and in human keratinocytes (*Lewis et al., 1994*; *Wheelock and Jensen, 1992*). Desmosome formation in keratinocytes requires junctional initiation by the classical cadherins, Ecad or P-cadherin (Pcad) (*Amagai et al., 1995*; *Michels et al., 2009*). Consequently, Ecad and Pcad deficient mice show defective desmosome assembly (*Tinkle et al., 2008*). However, the precise molecular mechanisms by which classical cadherins promote desmosome formation are unknown.

Since Ecad interacts laterally to form *cis* dimers on the same cell surface (*Harrison et al., 2011*) while Ecad molecules from opposing cells interact in a *trans* strand-swap dimer conformation (*Boggon et al., 2002*; *Parisini et al., 2007*; *Vendome et al., 2011*) and a *trans* X-dimer conformation (*Ciatto et al., 2010*; *Harrison et al., 2010*), we used mutants that specifically abolish either Ecad *trans* or *cis* interactions and tested their binding to either Dsg2 or Dsc2. We characterized these interactions at different stages of desmosome formation using an integrated structure/function analysis that combined single molecule force measurements of wild type (WT) and mutant cadherins with an atomic force microscope (AFM), super-resolution imaging of desmosomes in human keratinocytes and confocal fluorescence microscopy of Ecad-knockout, Pcad-knockdown mouse keratinocytes ($E^{KO}/P^{KD}$), transfected with WT and mutant cadherins. The data identify a novel $Ca^{2+}$-independent direct interaction between Ecad and Dsg2 that is mediated by a conserved Leu 175 on Ecad. Previous structural studies have shown that L175 mediates homophilic Ecad *cis* dimerization (*Harrison et al., 2011*). Our data suggests that desmosome assembly is initiated in two stages: a first stage that requires stable Ecad *trans*-homodimerization and a second stage characterized by the direct heterophilic binding between Ecad and Dsg2 that facilitates further desmosome assembly. The interactions between Ecad and Dsg2 are short-lived and as desmosomes mature, Dsg2 dissociates from Ecad and forms stable bonds with Dsc2 to mediate robust adhesion.

## Results

### Ecad interacts with Dsg2 to form a $Ca^{2+}$-independent heterodimer

We identified the binding partners for recombinant Dsc2, Dsg2 and Ecad using single molecule AFM force measurements. The complete extracellular region of Dsc2, Dsg2 and Ecad were expressed in mammalian cells and biotinylated at their C-terminus using a biotin ligase enzyme (Materials and methods). Identical concentrations of biotinylated cadherins were immobilized on AFM tips and glass coverslip (CS) substrates that were functionalized with polyethylene glycol (PEG) tethers and decorated with streptavidin protein (*Figure 1A and B*), (Materials and methods), (*Lowndes et al., 2014*; *Manibog et al., 2014*; *Rakshit et al., 2012*; *Sivasankar et al., 2009*; *Zhang et al., 2009*). Under similar experimental conditions, cadherin surface density was previously determined to be $65 \pm 18$ cadherins per $\mu m^2$, which corresponds to an average distance of 124 nm between neighboring cadherins (*Zhang et al., 2009*). Since the separation between neighboring cadherins is an order of magnitude larger than the radius of curvature of the AFM tip, the measured unbinding events correspond to the interaction of only a single cadherin immobilized on the surface and the AFM tip respectively. At the start of the experiment, the AFM cantilever and substrate were brought into contact to allow opposing cadherins to interact. The tip was then withdrawn from the substrate and the force required to rupture the adhesive complex was measured. Interaction of opposing cadherins resulted in unbinding events characterized by non-linear stretching of the PEG tethers (*Figure 1C*); PEG stretching served as a molecular fingerprint for single molecule unbinding since its extension under load has been extensively characterized (*Oesterhelt et al., 1999*). If the cadherins did not interact, no unbinding forces were measured (*Figure 1D*).

To determine the adhesive properties of different cadherins, we measured the binding probabilities of various combinations of Dsg2, Dsc2 and Ecad in the presence of $Ca^{2+}$ or in the presence of EGTA, a $Ca^{2+}$ chelating agent. We characterized the levels of nonspecific interactions in every experiment by measuring both the binding probability of a cadherin functionalized AFM cantilever and a substrate lacking cadherin and also the binding probability of an AFM cantilever lacking cadherin

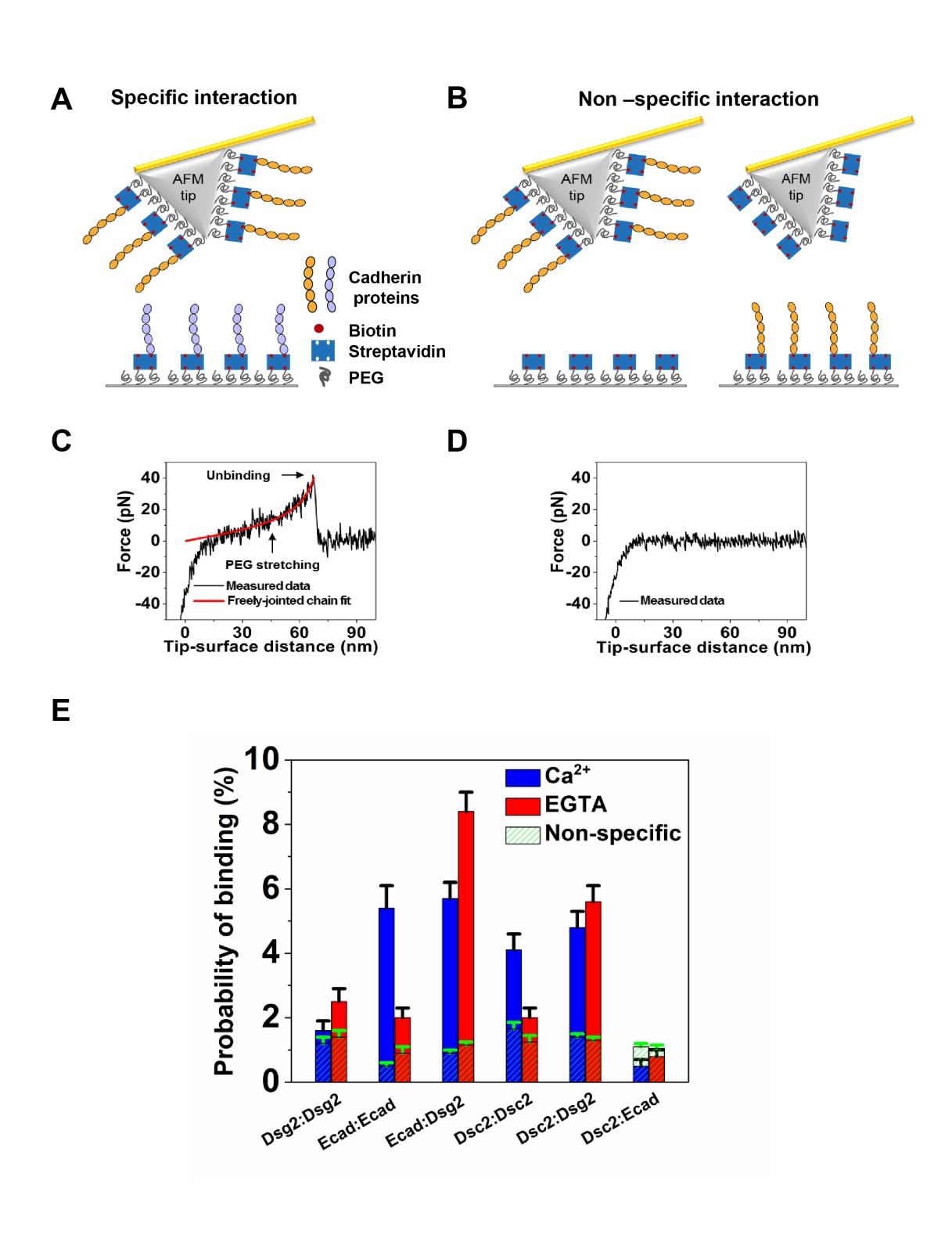

**Figure 1.** Ecad interacts with Dsg2 to form $Ca^{2+}$-independent dimers. (**A**) Schematic of specific interaction experiment. The AFM tip and substrate were functionalized with PEG linkers some of which were decorated with streptavidins. Biotinylated cadherin proteins were attached to streptavidin. (**B**) Schematic of nonspecific interaction experiment. The probability of interactions between the AFM tip functionalized with biotinylated cadherin proteins and the substrate lacking cadherins (left) and the binding probability of an AFM cantilever lacking cadherin and a substrate decorated with cadherins

*Figure 1 continued on next page*

*Figure 1 continued*

(right) was measured. Example force versus tip-surface distance traces showing (C) a single unbinding event with signature PEG stretching and (D) no interaction. (E) Specific binding probabilities for different combination of cadherins on the tip and substrate measured in $Ca^{2+}$ (blue) and in EGTA (red), a $Ca^{2+}$ chelator. Non-specific binding levels (hatched green) were determined from the average of measured binding probabilities between a cadherin functionalized AFM tip and a surface lacking cadherin and between an AFM tip lacking cadherin and surface functionalized with biotinylated cadherin proteins. Dsg2/Dsg2 data was from a total of 1666 ($Ca^{2+}$) and 1849 (EGTA) measurements; Ecad/Ecad data was from a total of 1052 ($Ca^{2+}$) and 2150 (EGTA) measurements; Ecad/Dsg2 data was from a total of 2215 ($Ca^{2+}$) and 2051 (EGTA) measurements; Dsc2/Dsc2 data was from a total of 1658 ($Ca^{2+}$) and 2025 (EGTA) measurements; Dsc2/Dsg2 data was from a total of 1850 ($Ca^{2+}$) and 2025 (EGTA) measurements; Dsc2/Ecad data was from a total of 2122 ($Ca^{2+}$) and 2098 (EGTA) measurements. Error bars are s.e. calculated using bootstrap with replacement.

DOI: https://doi.org/10.7554/eLife.37629.002

and a cadherin-functionalized substrate (*Figure 1B*); nonspecific binding probabilities (shown as green hatched bars in *Figure 1E*) are the average of both these sets of measurements. In agreement with our previous results (*Lowndes et al., 2014*), the probability of homophilic Dsg2 interaction in either the presence (1.6 ± 0.3%) or absence (2.5 ± 0.4%) of $Ca^{2+}$ was comparable to nonspecific adhesion (1.2 ± 0.2% in $Ca^{2+}$ and 1.4 ± 0.2% in EGTA). In contrast, Dsc2/Dsc2 showed a $Ca^{2+}$-dependent homophilic interaction (*Lowndes et al., 2014*), with a binding probability of 4.1 ± 0.5% in $Ca^{2+}$ and 2.0 ± 0.3% in EGTA (corresponding nonspecific binding was 1.6 ± 0.2% and 1.3 ± 0.2% in $Ca^{2+}$ and EGTA, respectively). Heterophilic interactions between Dsc2 (on AFM tip) and Dsg2 (on CS) were $Ca^{2+}$-independent (*Lowndes et al., 2014*), with binding probabilities of 4.8 ± 0.5% in $Ca^{2+}$ and 5.6 ± 0.5% in EGTA (nonspecific binding levels were 1.4 ± 0.1% and 1.3 ± 0.1% in $Ca^{2+}$ and EGTA, respectively). As expected, Ecad also showed a $Ca^{2+}$-dependent homophilic interaction with a binding probability of 5.4 ± 0.7% in $Ca^{2+}$ and 2.0 ± 0.3% in EGTA (corresponding nonspecific binding was 0.5 ± 0.1% in $Ca^{2+}$ and 0.9 ± 0.3% in EGTA) (*Figure 1E*).

Surprisingly, we also measured $Ca^{2+}$-independent heterophilic interactions between Dsg2 (on CS) and Ecad (on AFM tip) with binding probabilities of 5.7 ± 0.5% and 8.4 ± 0.6% in the presence of $Ca^{2+}$ and EGTA respectively (nonspecific binding: 0.9 ± 0.1% in $Ca^{2+}$ and 1.1 ± 0.1% in EGTA). In contrast, Dsc2 on AFM tip and Ecad on the CS did not show any heterophilic binding either in $Ca^{2+}$ or in EGTA; while a binding probability of 0.5 ± 0.2%, comparable to nonspecific adhesion of 1.1 ± 0.1%, was measured in $Ca^{2+}$, the binding frequency of Dsc2/Ecad interactions in EGTA was 0.8 ± 0.2% similar to nonspecific binding of 1.1 ± 0.1%. Our binding probability measurements thus demonstrate that Dsg2 and Ecad form a $Ca^{2+}$-independent heterophilic dimer while Dsc2 does not bind heterophilically to Ecad.

## Ecad/Dsg2 and Dsc2/Dsc2 dimers have lower lifetimes than Dsc2/Dsg2 dimers

Next, we compared the dissociation rates for Dsc2/Dsc2, Dsc2/Dsg2 and Ecad/Dsg2 dimers, in the presence of $Ca^{2+}$, using single molecule dynamic force spectroscopy (DFS). The cadherins were immobilized on the AFM tip and substrate as described above and the surface density of the protein was empirically adjusted such that the binding probability was ~6%. Under these conditions, Poisson statistics predicts that more than 97% of measured events are from the rupture of single bonds. The single molecule unbinding events, which were characterized by the non-linear stretching of the PEG tethers (*Figure 1C*), were fit to an extended freely-jointed chain model (*Oesterhelt et al., 1999*) using a total least squares fitting protocol. Specific unbinding events were unambiguously identified since they occurred at a distance corresponding to the contour length of two PEG tethers; only specific unbinding events were used in further analysis (see Materials and methods). The measurements were repeated several thousand times at six different rates of application of force (loading rates) and at different positions of the substrate.

The most probable unbinding force at the different loading rates were fit to the Bell-Evans model (*Bell, 1978*; *Evans and Ritchie, 1997*) to measure the intrinsic off-rate under zero force, $k^0_{off}$ and the width of energy barrier that inhibit protein dissociation, $x_\beta$ (*Figure 2A,B,C*). We used cluster analysis to group the single molecule unbinding events for fitting (*Yen and Sivasankar, 2018*). We have previously shown that a K-means clustering algorithm greatly improves the estimation of kinetic parameters in DFS (*Yen and Sivasankar, 2018*). This analysis showed that the off-rate of Dsc2/Dsc2 dimers (*Figure 2A*) and Dsg2/Ecad dimers (*Figure 2C*) were comparable with a $k^0_{off}$ of 1.26 s$^{-1}$ and

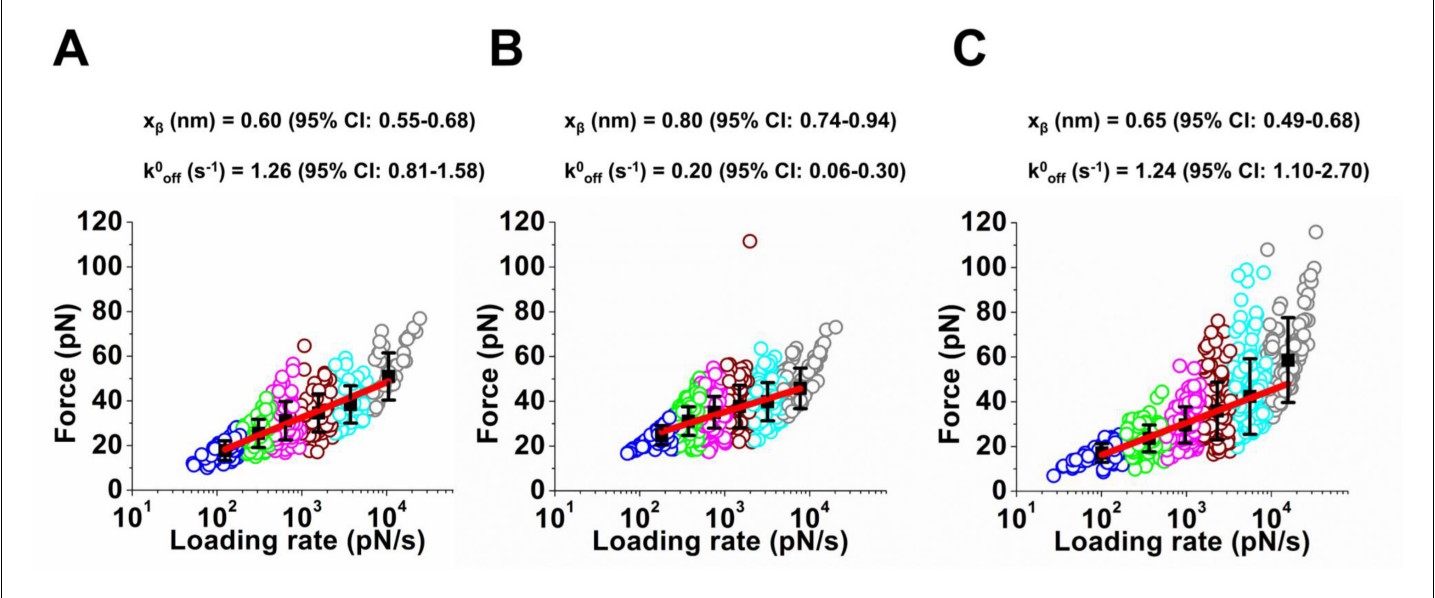

**Figure 2.** Lifetimes of the Ecad/Dsg2 dimer and the Dsc2/Dsc2 dimer are shorter than the lifetime of the Dsg2/Dsc2 complex. Loading rates of the rupture events measured in $Ca^{2+}$ at six different pulling velocities were grouped using K-means clustering method. Each clustered loading rate is shown by a different color, with each circle represent a single rupture event. The mean force and mean loading rates (black filled squares) for the groups were fit to Bell-Evans model (red line) using a nonlinear least-squares fitting with bisquare weights. Fits yielded the intrinsic off-rate ($k^0_{off}$) and the width of the transition energy barrier ($x_\beta$). Error bars in force correspond to standard deviation. 95% confidence interval (CI) was calculated using bootstrap with replacement. Analysis shown for (A) Dsc2/Dsc2 (B) Dsc2/Dsg2 and (C) Ecad/Dsg2. The data shown in panels A, B and C correspond to 415 events, 988 events, and 725 events respectively.

DOI: https://doi.org/10.7554/eLife.37629.003

1.24 $s^{-1}$ respectively. In contrast, the lifetime of the Dsc2/Dsg2 dimer (*Figure 2B*) was 6x longer with a smaller $k^0_{off}$ of 0.20 $s^{-1}$, demonstrating that the Dsg2/Dsc2 dimer was more stable than either the Dsc2/Dsc2 or the Dsg2/Ecad. These measurements suggest that, upon dissociation of Dsg2/Ecad and Dsc2/Dsc2 complexes, the free Dsg2 and Dsc2 would preferentially bind. In agreement with our data, recent solution binding affinity measurements of desmosomal cadherins have shown heterophilic interactions are stronger than homophilic binding (*Harrison et al., 2016*).

## Ecad is present in nascent desmosomes but not in mature desmosomes

Next, we used structured illumination microscopy (SIM) to test for the presence of Ecad at different stages of desmosome assembly in human keratinocytes. Keratinocytes were first cultured in a medium containing a low concentration of $Ca^{2+}$ ions not conducive for desmosome formation (100 µM $Ca^{2+}$) and the $Ca^{2+}$ concentration was then increased to trigger desmosome assembly (550 µM $Ca^{2+}$), (Materials and methods). At three time points following the $Ca^{2+}$ switch (1 hr, 3 hr and 18 hr), keratinocytes were fixed and immunostained for Dsg2, Ecad and Desmoplakin (DP), a protein that links desmosomal proteins to the intermediate filament cytoskeleton. Since DP is an obligate desmosomal protein, its distribution allowed us to identify individual desmosomes on the keratinocyte surface with desmosomal junctions defined by regions of parallel DP 'railroad tracks' (*Stahley et al., 2016a*; *Stahley et al., 2016b*) (*Figure 3A,B*).

Comparison of relative Dsg2 and Ecad levels contained within the DP railroad tracks demonstrated that Ecad levels were enriched in nascent desmosomes, with relative levels decreasing as desmosomes matured (*Figure 3B,C*). Compared to Ecad levels in nascent desmosomes, Ecad fluorescence intensity decreased by 50% in mature desmosomes (*Figure 3C*). In contrast, the relative levels of Dsg2 stayed constant at all time points examined (*Figure 3C*). Importantly, relative levels of Ecad along the entire cell border did not change with time (*Figure 3—figure supplement 1*), confirming that the Ecad enrichment within DP railroad tracks at early time points is specific to desmosomal regions of the membrane, and was significant beyond that expected by random chance or

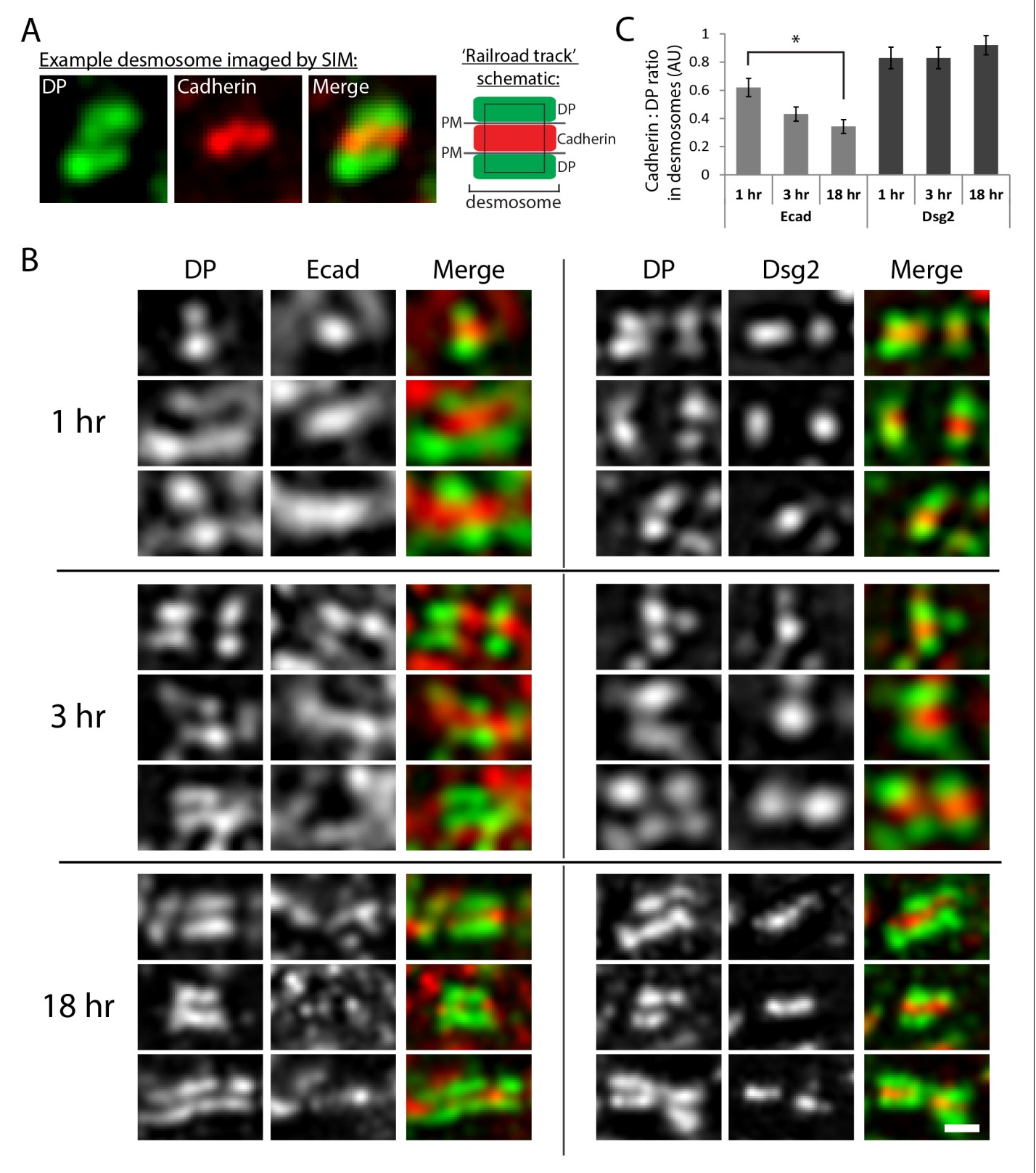

**Figure 3.** Ecad and Dsg2 are both localized in nascent desmosomes. (**A**) Analysis of cadherin localization within desmosomes. Structured illumination microscopy (SIM) is able to resolve the distance from plaque to plaque when desmosomes are stained with a C-terminal DP antibody and an N-terminal cadherin antibody, as shown in the example SIM image (*Figure 3A*). Desmosomes were defined by regions of parallel DP staining, or 'railroad tracks'. *Figure 3 continued on next page*

Figure 3 continued

DP (green) and cadherin (either Ecad or Dsg2, red) fluorescence intensity were measured within the desmosome region of interest (black rectangle). (B) Representative images of desmosomal regions in human keratinocytes cultured in high $Ca^{2+}$ media for 1, 3 or 18 hr as indicated. Images are oriented with cell border horizontal. Scale bar, 0.5 μm. (C) Quantification of cadherin (Ecad or Dsg2) levels relative to DP in desmosomes at different time points after initiation of desmosome assembly with high $Ca^{2+}$ culture conditions. AU, arbitrary units. Means ± SE, n = 25 desmosomes, *p<0.05.
DOI: https://doi.org/10.7554/eLife.37629.004
The following figure supplement is available for figure 3:

**Figure supplement 1.** Relative Ecad levels remain unchanged over a calcium switch time-course.
DOI: https://doi.org/10.7554/eLife.37629.005

because of changes in E-cadherin localization unrelated to junction maturation. Taken together, these data indicate that prior to desmosome maturation, Ecad is significantly enriched in nascent desmosomes.

## Leu 175 mediates Ecad and Dsg2 interactions

Next, we proceeded to use single molecule AFM measurements to determine the precise molecular interactions that mediate Ecad/Dsg2 binding. We used mutants that specifically abolish either Ecad *trans* or *cis* interactions and tested their binding to Dsg2. Structural studies show that *trans* strand-swap dimer formation can be eliminated by mutating a conserved Trp2 to Ala (W2A) (*Harrison et al., 2010*). Similarly, mutating a conserved Lys14 to Glu (K14E) eliminates a key salt-bridge in the X-dimer interface and abolishes X-dimer formation (*Harrison et al., 2010*). We therefore tested the binding between the Ecad W2A-K14E double mutant (DM) and Dsg2. In agreement with previous results (*Harrison et al., 2010*; *Rakshit et al., 2012*), we confirmed that the DM cannot interact homophilically; the binding interaction between opposing DMs (1.4 ± 0.3% in $Ca^{2+}$ and 1.8 ± 0.3% in EGTA) were comparable to nonspecific binding in $Ca^{2+}$(1.4 ± 0.1%) and in EGTA (1.1 ± 0.1%) (*Figure 4*). However, when we measured the interactions between the DM on AFM tip and Dsg2 on CS, our data showed that the DM interacts with Dsg2 in a $Ca^{2+}$-independent heterophilic fashion; while the DM/Dsg2 binding probability in $Ca^{2+}$ and in EGTA was 6.1 ± 0.5% and 6.9 ± 0.5% respectively, the non-specific interaction in $Ca^{2+}$ and EGTA was 1.3 ± 0.1% and 1.2 ± 0.1% respectively (*Figure 4*). This demonstrates that the Ecad/Dsg2 binding interface is different from the previously established interface for Ecad *trans* dimerization.

Next, we tested whether Ecad interacts with Dsg2 via its *cis* dimer interface. Since previous studies had shown that mutating a conserved Leu 175 to Asp (L175D) eliminates Ecad *cis* dimerization (*Harrison et al., 2011*), we measured the interaction of this Ecad *cis* mutant (CM) and Dsg2. First, we confirmed that CM was functional by measuring its *trans* binding probability. As shown previously (*Harrison et al., 2011*), our data confirmed that the CM forms $Ca^{2+}$-dependent *trans* dimers; the binding probabilities in $Ca^{2+}$ and in EGTA were 4.1 ± 0.5% and 0.8 ± 0.2% respectively (corresponding nonspecific binding probabilities in $Ca^{2+}$ and in EGTA were 1.2 ± 0.2% and 1.4 ± 0.2% respectively) (*Figure 4*). However, the interaction of CM (on CS) and Dsg2 (on AFM tip) was comparable to the measured nonspecific binding; we measured binding probabilities of 1.8 ± 0.3% in $Ca^{2+}$ and 1.5 ± 0.3% in EGTA which were similar to measured nonspecific binding levels of 1.2 ± 0.1% in $Ca^{2+}$ and 1.4 ± 0.1% in EGTA (*Figure 4*). Previous simulations (*Wu et al., 2010*; *Wu et al., 2011*) and single molecule FRET experiment (*Zhang et al., 2009*) have shown that *cis* homo-dimerization of Ecad requires prior *trans* dimer formation. Thus, the failure to form CM/Dsg2 interactions is not due to the abrogation of Ecad *cis* dimers. Taken together, these experiments show that Ecad/Dsg2 dimerization is mediated by L175, which also mediates *cis* homo-dimerization in Ecad.

## Ecad L175 is essential for efficient intercellular Dsg2 recruitment and desmosome assembly

To test whether the Ecad L175D mutation can impede recruitment of Dsg2 in keratinocytes, we expressed either full-length Ecad WT or mutants in $E^{KO}/P^{KD}$ keratinocytes and analyzed Dsg2 recruitment to sites of intercellular contacts. We have previously shown that the $E^{KO}/P^{KD}$ keratinocytes are unable to assemble adherens junction (AJs) and desmosomes due to the loss of all classical cadherins (*Michels et al., 2009*). As we wanted to assess the ability of Ecad mutants to recruit desmosomal

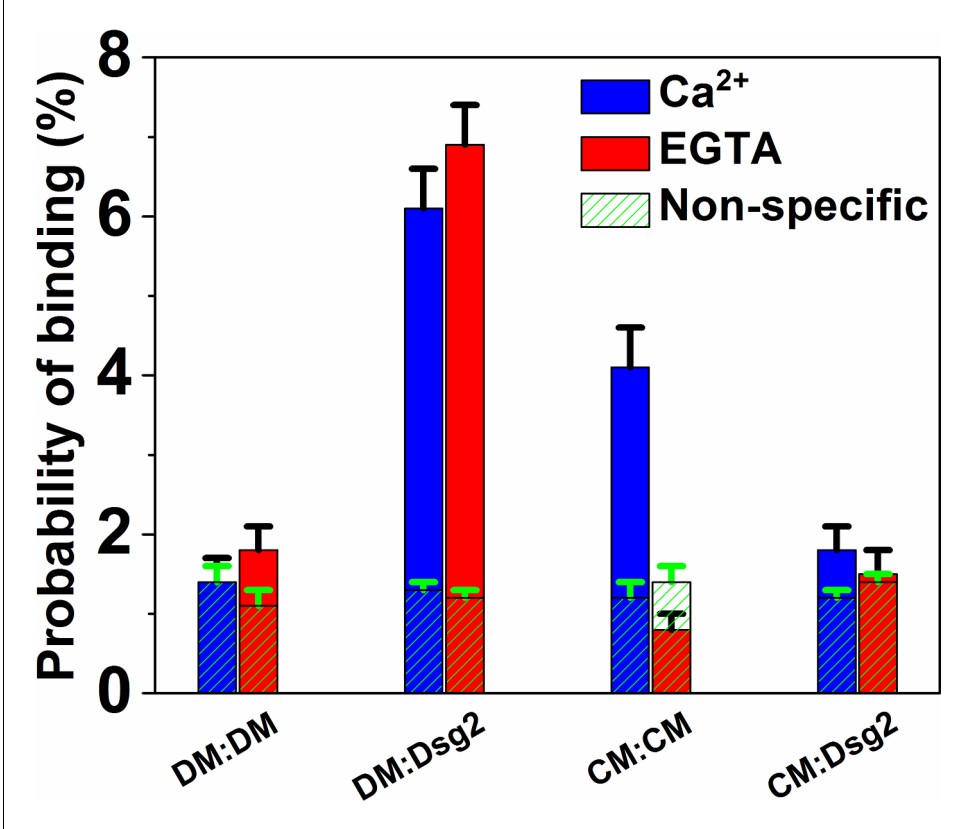

**Figure 4.** Ecad interacts with Dsg2 via Leu 175. Homophilic binding probability of Ecad W2A-K14E double mutant (DM); heterophilic binding probability of DM and Dsg2; homophilic binding probability of Ecad L175D *cis* dimer mutant (CM); and the heterophilic binding probability of CM and Dsg2 was measured in $Ca^{2+}$ (blue) and in EGTA (red). Nonspecific binding probabilities determined from the average of measured binding probabilities between a cadherin functionalized AFM tip and a surface lacking cadherin and between an AFM tip lacking cadherin and surface functionalized with biotinylated cadherins are shown in shaded green. DM/DM data was from a total of 1898 ($Ca^{2+}$) and 2122 (EGTA) measurements; DM/Dsg2 data was from a total of 2150 ($Ca^{2+}$) and 2009 (EGTA) measurements; CM/CM data was from a total of 1970 ($Ca^{2+}$) and 1906 (EGTA) measurements; CM/Dsg2 data was from a total of 2027 ($Ca^{2+}$) and 2122 (EGTA) measurements. Error bars are s.e. calculated using bootstrap with replacement.

DOI: https://doi.org/10.7554/eLife.37629.006

proteins early during junction formation, we performed confocal microscopy on keratinocytes immunostained for Ecad, Dsg2 and DP, at three time points following the $Ca^{2+}$ switch (3 hr, 6 hr and 18 hr, *Figure 5*, Materials and methods). The ability of transfected keratinocytes to form AJs was first assessed by the formation of 'zipper-like' patterns of Ecad at intercellular contacts. We have previously shown that these zippers represent early AJs and recruit AJ marker proteins like vinculin (*Rübsam et al., 2017b*). Only intercellular interfaces with AJ zippers were examined for their ability to recruit Dsg2 and DP to these contacts (*Figure 5—figure supplement 1A*). Importantly, we never observed any enrichment of desmosomal components at intercellular contacts in the absence of AJ zippers (*Michels et al., 2009*).

In agreement with previous results (*Vasioukhin et al., 2000*), our data showed that 3 hr after the $Ca^{2+}$ switch, 93% of Ecad WT was enriched in zipper-like early AJs at sites of intercellular contacts. In contrast only 48% of Ecad-L175D transfected keratinocytes formed AJ zippers, likely due to impaired Ecad *cis*-dimer formation (*Figure 5A,B*) (*Harrison et al., 2011*). At these early time points following the $Ca^{2+}$ switch, 66% of these Ecad WT zipper contacts were positive for Dsg2. Junctional localization of Ecad and Dsg2 increased to 97%, 18 hr after allowing cells to engage in $Ca^{2+}$-dependent intercellular adhesion (*Figure 5A,C*). In contrast, only 39% of Ecad-L175D induced zipper

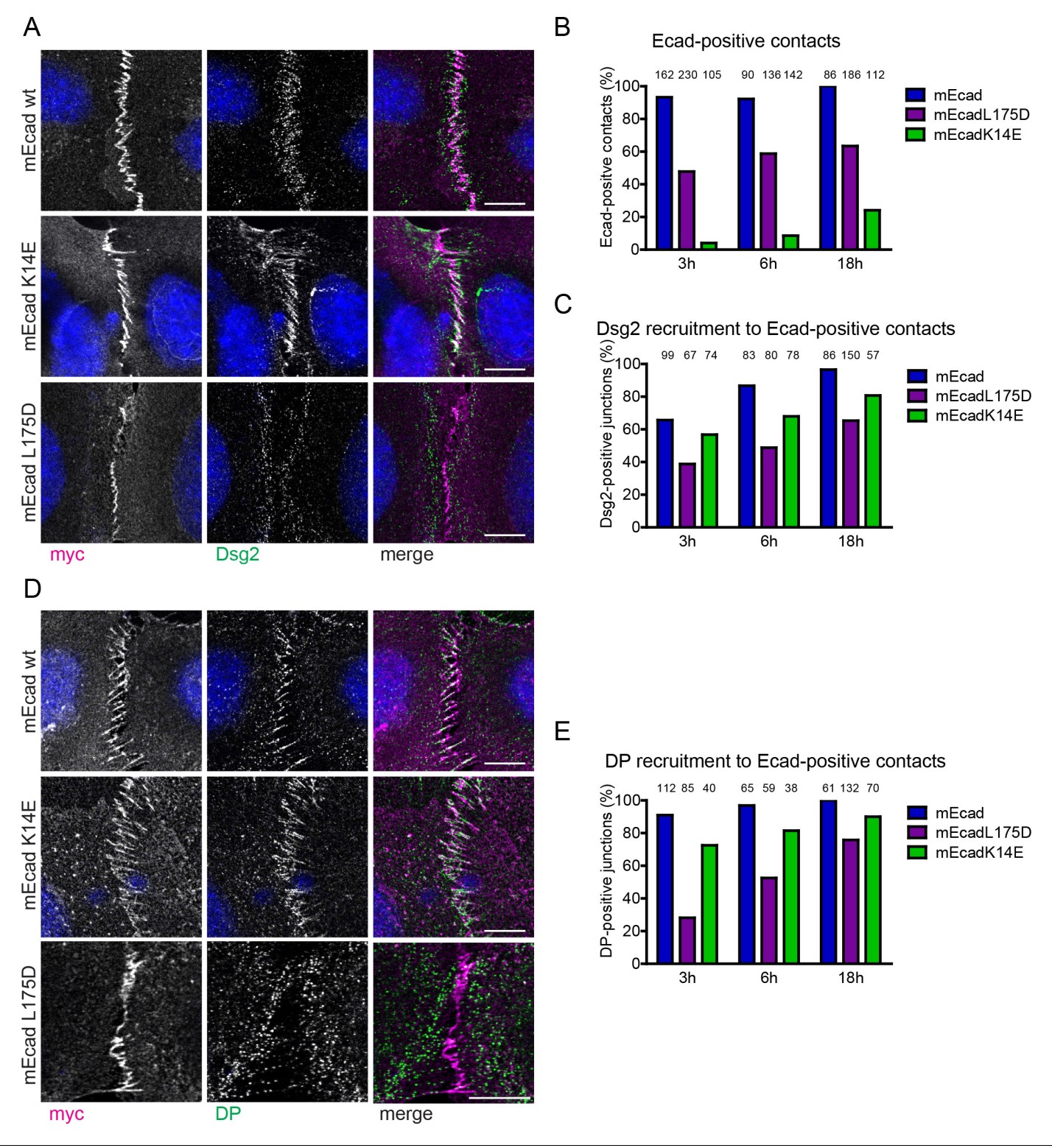

**Figure 5.** Ecad-L175 promotes desmosome assembly in cells. (**A**) Immunofluorescence analysis for transfected WT or mutant Ecad (myc) and Dsg2, 6 hr after allowing de novo junction formation in Ecad[KO]/Pcad[KD] keratinocytes. Note decreased localization of Dsg2 at intercellular contacts formed by Ecad-L175D. (**B**) Quantification of adherens junction (AJ) formation at intercellular contacts, judged by zipper-like enrichment of Ecad constructs. (**C**) Quantification of Dsg2 recruitment to AJ formed by WT or mutant Ecad. (**D**) Immunofluorescence analysis for transfected WT or mutant Ecad (myc) and desmoplakin (DP), 6 hr after allowing de novo junction formation in Ecad[KO]/Pcad[KD] keratinocytes. Note decreased localization of DP at intercellular

*Figure 5 continued on next page*

*Figure 5 continued*

contacts formed by Ecad-L175D. (E) Quantification of DP recruitment to AJ formed by WT or mutant Ecad. Numbers of quantified junctions (interface of two contacting, transfected cells) are shown above each bar and are derived from at least three independent experiments. Scale bar: 10 μm.

DOI: https://doi.org/10.7554/eLife.37629.007

The following figure supplement is available for figure 5:

**Figure supplement 1.** Impaired junction formation in Ecad-K14E, Ecad-L175D, and Ecad-W2A/K14E (DM) mutants.

DOI: https://doi.org/10.7554/eLife.37629.008

contacts showed Dsg2 recruitment 3 hr after switching to high $Ca^{2+}$ (*Figure 5A,C*), which increased to 65% at 18 hr, confirming that a direct interaction between Ecad and Dsg2 is required for efficient recruitment of Dsg2 to early intercellular contacts.

To confirm that the observed effect of the L175D mutation was not due to hindered AJ formation, we transfected the $E^{KO}/P^{KD}$ keratinocytes with the full-length Ecad-K14E mutant that abolishes X-dimer formation and traps Ecad in a strand-swap dimer conformation (*Harrison et al., 2010*; *Rakshit et al., 2012*). Our data showed that only 4% of the transfected, contacting cells formed zippers at early (3 hr) time-points after switching to high $Ca^{2+}$ with only 24% of the transfected contacting cells showing zippers at late (18 hr) time points (*Figure 5A,B*, *Figure 5—figure supplement 1*), confirming that K14 is essential for effective AJ formation (*Hong et al., 2011*), and thus intercellular contact establishment. However, the few Ecad-K14E AJs that were formed, recruited Dsg2 more efficiently than Ecad-L175D (*Figure 5A,C*, *Figure 5—figure supplement 1*). This result confirmed that delayed recruitment of Dsg2 to intercellular contacts was not the result of the inability of Ecad-L175D to efficiently form AJs but rather due to absence of direct interaction between Ecad and Dsg2.

That there is no co-localization of DP and Ecad in the absence of zippers (*Figure 5—figure supplement 1*) suggests that Ecad *trans* interactions precede Ecad/Dsg2 interactions. This conclusion is further strengthened by our finding that DP was also not recruited upon transfection of the full-length Ecad-DM (W2A-K14E double mutant, *Figure 5—figure supplement 1*) in $E^{KO}/P^{KD}$ keratinocytes, which abolishes Ecad *trans* adhesion, and thus zipper formation.

Finally, to examine whether L175D only interferes with Dsg2 recruitment or more generally delays desmosome assembly, we also examined the ability of Ecad-L175D to hinder DP recruitment. In $E^{KO}/P^{KD}$ keratinocytes transfected with Ecad WT, DP was enriched in 91% of Ecad-positive intercellular contacts after 3 hr in high $Ca^{2+}$ and in 100% of contacts after 18 hr in high $Ca^{2+}$ (*Figure 5D,E*). In contrast, only 28% of Ecad-L175D established intercellular contacts were positive for DP after 3 hr, which increased to 72% after 18 hr. Importantly, after 3 hr, 73% of Ecad-K14E mutant contacts were DP positive, despite widespread defects in AJ formation (*Figure 5B,D,E*). Taken together these results confirm that amino acid L175 in Ecad mediates Dsg2 interactions and facilitates early desmosome complex formation in cells.

## Discussion

Here, we integrate in vitro single molecule and cellular structure-function experiments to identify two critical events that initiate and promote efficient desmosome assembly: (i) stable *trans*-homodimerization of Ecad, and (ii) the direct heterophilic binding of Ecad and Dsg2 ectodomains. Our data demonstrates that desmosome assembly is initiated at sites of Ecad *trans* homodimerization. We also show that Ecad and Dsg2 bind via a conserved Leu 175 on the Ecad *cis* binding interface and form short-lived heterophilic complexes that localize to early desmosomes and efficiently recruit desmosomal proteins to sites of intercellular contact formation.

Previously, biochemical analysis identified Ecad/Dsg and Dsc/Dsg complexes upon removing $Ca^{2+}$ from the cell culture (*Troyanovsky et al., 1999*). Our data builds upon these previous findings by showing a direct interaction between Ecad and Dsg that also occurs under conditions that promote intercellular adhesions and by identifying the Ecad interface responsible for the interaction. We also demonstrate that in a physiological setting, L175 on Ecad is important for recruitment of Dsg2 and DP and for efficient desmosome formation. It has also been suggested that cross-talk between classical and desmosomal cadherins is mediated by plakoglobin, a cytoplasmic signaling protein found in both AJs and desmosomes (*Lewis et al., 1997*). While we did not test the role of

plakoglobin in desmosome formation, our data indicates that a direct physical interaction between Ecad and Dsg2 ectodomains is critical for timely desmosome assembly.

It is important to point out that Ecad/Dsg interactions reported here may not be unique to Dsg2 but may occur with other Dsg isoforms as well. In keratinocytes, Dsg3 is essential for desmosome assembly (*Hartlieb et al., 2013*; *Koch et al., 1997*) and a previous fluorescence co-localization and co-immunoprecipitation study reported interactions between Ecad and Dsg3 (*Tsang et al., 2010*). Since the $E^{KO}/P^{KD}$ keratinocytes transfected with Ecad-L175D also express Dsg3, it is possible that poor junctional DP enrichment seen at early time-points, which was more impaired than Dsg2 recruitment, may result from the inability of Ecad-L175 to bind Dsg3.

Desmosome assembly is $Ca^{2+}$ dependent and previous models using Dsc2 and Dsg2 propose that desmosomal cadherins assemble in two phases (*Burdett and Sullivan, 2002*; *Lowndes et al., 2014*). The first phase of assembly is believed to involve the clustering of Dsc into nucleation sites. Since Dsc2 homo-dimerization is $Ca^{2+}$ dependent (*Figure 1E*), it follows that desmosome assembly requires $Ca^{2+}$, even though Dsc2/Dsg2 heterodimers are $Ca^{2+}$ independent. In the second phase of desmosome formation, Dsg2 is recruited to the clustered Dsc2 nucleation sites through an unknown mechanism that is $Ca^{2+}$ and W2 independent and that relies on heterophilic interactions with other proteins (*Lowndes et al., 2014*). Our data suggests that Ecad mediates $Ca^{2+}$ independent recruitment of Dsg2, and perhaps other Dsgs, to nascent Dsc clusters. Previous studies have also shown that initial cell-cell contacts subsequently trigger multiple phases of DP recruitment to finally assemble desmosomes (*Godsel et al., 2005*).

Due to the transient nature of Ecad/Dsg2 interactions, desmosomal proteins recruited to junctions are expected to rapidly segregate from Ecad. Consequently, images of intercellular interfaces are expected to have both overlapping and separate Ecad and desmosomal protein staining patterns. In agreement, although Dsg2 and DP show some co-localization at Ecad-positive zippers (*Figure 5*), significant non-overlapping junctional staining is observed, even within the first 3 hr. This suggests that AJs and desmosomes rapidly segregate into distinct intercellular junctions.

Although our biophysical experiments demonstrate that Ecad-DM directly interact with Dsg2, imaging of keratinocytes transfected with Ecad-K14E and Ecad-DM show that stable *trans* Ecad homodimerization is required for DP and Dsg2 recruitment (*Figure 5—figure supplement 1*). While the K14E mutants are strongly impaired in AJ zipper formation, they only localize with Dsg2 or DP when AJ zippers are present. Similarly, cells expressing Ecad-DM which do not form AJs, also do not recruit DP or Dsg2. These results indicate that Ecad *trans* binding serves as the initial spatial cue for the subsequent recruitment of Dsg2 and consequently for desmosome assembly. These results are thus in line with the observation that combined loss of both E-, and Pcad, the two main classical cadherins expressed in keratinocytes, prevents desmosome assembly (*Michels et al., 2009*). Mechanistically, the cross talk between Ecad *trans* dimerization and Ecad/Dsg2 binding may be analogous to Ecad *cis* homodimerization via L175, which occurs only when conformational entropy is reduced by prior Ecad *trans* binding (*Wu et al., 2010*; *Wu et al., 2011*). Alternatively, Ecad *trans* dimers may be needed to bring opposing cells closer together to initiate desmosome formation. In support of this possibility, a previous study has suggested that when opposing HeLa cell membranes are brought into close proximity, by the interaction of protein zero neuronal adhesion molecules, desmosomes are formed (*Doyle et al., 1995*).

Based on these previous studies and our new data, we propose a model (*Figure 6*) whereby desmosome assembly is facilitated both by the direct *cis* interactions of classical cadherins and Dsg and the *trans* binding of opposing classical cadherins. In our model, the *trans* homodimerization of Ecad from opposing cells serves as a spatial cue to coordinate desmosome assembly. Ecad subsequently forms *cis* dimer complexes with Dsg though the results presented here do not rule out Ecad/Dsg binding in a *trans* conformation. Furthermore, the data suggest that once localized within the native desmosome, Dsg dissociates from Ecad and binds to Dsc to form mature desmosomes. Since the Dsc2/Dsg2 complex has a longer lifetime than both the Ecad/Dsg2 complex and Dsc2/Dsc2 complex, the Dsg/Dsc interactions likely promote robust cell-cell adhesion and permit the mature desmosome to withstand mechanical force.

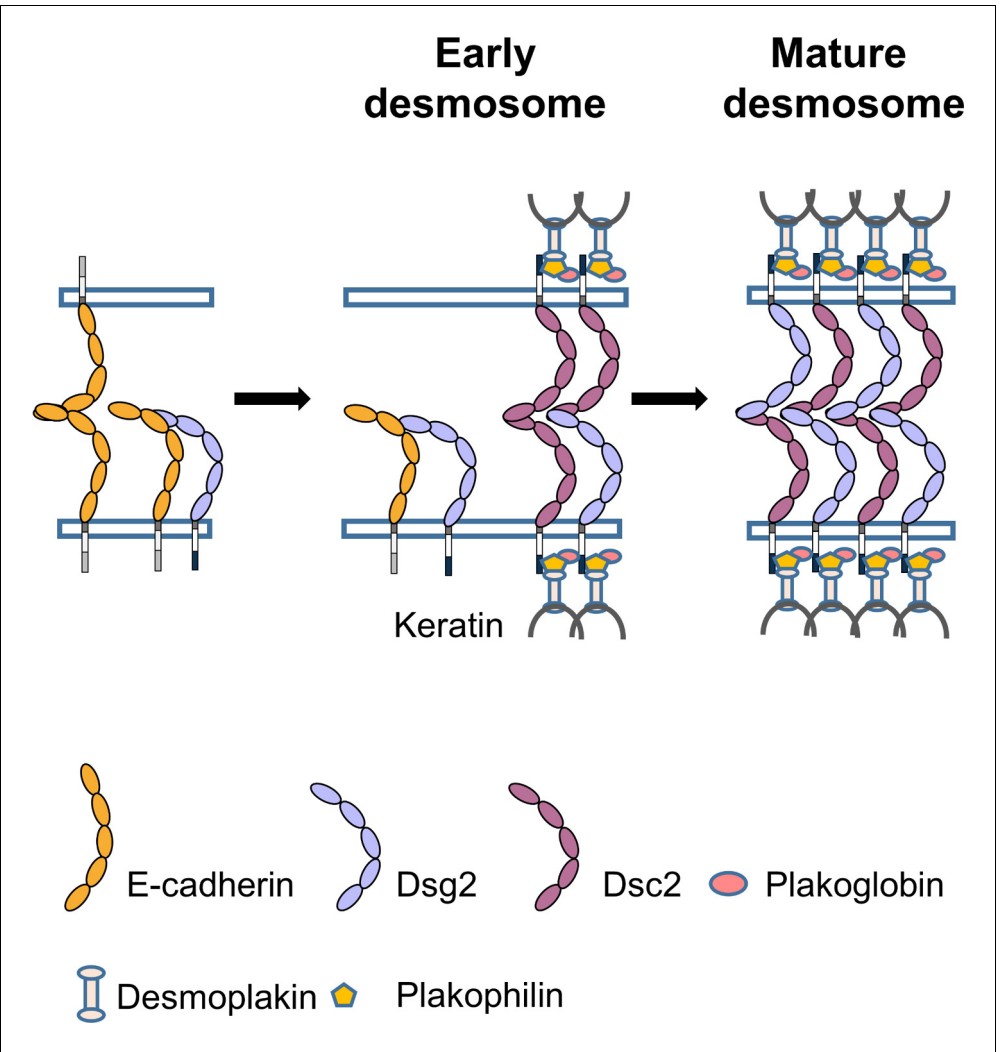

**Figure 6.** Proposed model for the role of Ecad in desmosome assembly. Ecad (orange) interacts with Dsg2 (light blue) to form a low-lifetime *cis* heterodimer. Formation of Ecad/Dsg2 complex requires prior Ecad *trans* homodimerization. The Ecad/Dsg2 complex is incorporated in the nascent desmosome which also contains low-lifetime Dsc2/Dsc2 dimers (purple). As the desmosome matures, the Ecad/Dsg2 heterodimers and Dsc2/Dsc2 homodimers dissociate. The Dsg2 and Dsc2 interact to form a robust, high-lifetime, *trans* adhesive complex.
DOI: https://doi.org/10.7554/eLife.37629.009

## Materials and methods

### Purification of cadherin ectodomains

The generation of HEK293T cells stably expressing Ecad WT and Ecad W2A-K14E fused to a C-terminal Avi-tag and His-tag have been described previously (*Rakshit et al., 2012*; *Zhang et al., 2009*). Plasmids for full length Dsc2 and Dsg2 ectodomains fused to a C-terminal Avi-tag and His-tag were a kind gift from Prof. W. James Nelson (Stanford University) while HEK293T cells stably expressing full-length Ecad mutant L175D fused to a C-terminal Avi-tag and His-tag were a kind gift from Dr. Yunxiang Zhang (Stanford University). As described previously (*Lowndes et al., 2014*), the Dsc2 and Dsg2 plasmids were transiently transfected into HEK 293 T cells using lipofactamine 2000 (Life Technologies) following the manufacturer's protocol. Two days post transfection, the conditioned media was collected for protein purification. The transfected cells expressing WT/mutant Ecads were grown to confluence in DMEM containing 10% FBS and 200 µg/ml of Genecitin (G418; Invitrogen) and

exchanged into serum free DMEM with 400 µg/ml of Genecitin. Conditioned media was collected 4 days after media exchange.

Purification and biotinylation of His-tagged Dsc2, Dsg2, Ecad WT, W2A-K14E, and L175D followed a protocol that has been described previously (*Lowndes et al., 2014*; *Rakshit et al., 2012*; *Zhang et al., 2009*). Media containing cadherin was incubated overnight, at 4°C, with Ni NTA agarose beads (Qiagen). The beads were loaded onto a glass chromatography column (BioRad) and washed with buffer at pH 7.5 (20 mM $NaH_2PO_4$, 500 mM NaCl and 1 mM $CaCl_2$) containing 50 mM imidazole. The bound protein was eluted with the same buffer containing 250 mM imidazole. Following purification, the protein was exchanged into a pH 7.5 buffer containing 25 mM Hepes, 5 mM NaCl, and 1 mM $CaCl_2$ and biotinylated with BirA enzyme (BirA500 kit; Avidity). After biotinylation for 1 hr at 30°C, free biotins were removed using either a spin column (Millipore) or superdex 200 10/300 GL column.

## Single molecule AFM force measurements

Purified cadherins were immobilized on coverslips (CS) and AFM cantilevers (Olympus, model TR400PSA) using a previously described method (*Manibog et al., 2016*). Briefly, the CS and cantilevers were cleaned with 25% $H_2O_2$:75% $H_2SO_4$ and washed with DI water. The CS was then cleaned with 1 M KOH and washed with DI water. Both the CS and cantilevers were washed with acetone and functionalized using 2% (v/v) 3-aminopropyltriethoxysilane (Sigma) dissolved in acetone. Next, N-hydroxysuccinimide ester functionalized PEG spacers (MW 5000, Lysan Bio) were covalently attached to the silanized AFM tip and coverslip; 7% of the PEG spacers were decorated with biotin groups. Prior to a measurement, the functionalized AFM cantilever and coverslip were incubated overnight with BSA (1 mg/ml) to further reduce non-specific binding. The tip and surface were then incubated with 0.1 mg/ml streptavidin for 30 min and biotinylated cadherins were attached to the streptavidin. Finally, the surfaces were incubated with 0.02 mg/ml biotin for 10 min to block the free biotin binding sites on streptavidin.

Force measurements were performed using an Agilent 5500 AFM with a closed loop scanner. The spring constants of the cantilevers were measured using the thermal fluctuation method (*Hutter and Bechhoefer, 1993*). All the experiments were performed in a pH 7.5 buffer containing 10 mM Tris-HCl, 100 mM NaCl and 10 mM KCl with either 2.5 mM $Ca^{2+}$ or 2 mM EGTA. The region of PEG stretching in each unbinding force curve was fit to an extended freely jointed chain model using a total least squares fitting protocol. The contour length $L_c$ of the PEG tethers was determined from the fits. The histogram of $L_c$ for each experiment was fit to a Gaussian distribution and only force curves that had an $L_c$ within one standard deviation from the center were accepted for further analysis. Loading rates were calculated as described elsewhere (*Ray et al., 2007*). K-means clustering method was used to group loading rates (*Yen and Sivasankar, 2018*). Mean force $F^*$ and mean loading rate $r_f$ were calculated for each group and plots of $F^*$ vs. $r_f$ were fit using nonlinear least-squares fitting with bisquare weights to the Bell-Evans model (*Bell, 1978*; *Evans and Ritchie, 1997*). Confidence intervals (CIs) for $k_{off}^0$ and $x_\beta$ were determined using bootstrap-with-replacement, as described previously (*Yen et al., 2016*).

## SIM imaging and analysis of cadherin localization within desmosomes

Primary human keratinocytes (HKs, passage 2) were isolated from neonatal foreskin as previously described (*Calkins et al., 2006*) and cultured in KBM-Gold basal medium (100 µM calcium) supplemented with KGM-Gold Single-Quot Kit (Lonza, Walkersville, MD). HKs were cultured to 70% confluence on glass coverslips, switched to 550 µM calcium to induce junction assembly for the various time points indicated and then processed for structured illumination microscopy (SIM) as described below.

HKs were fixed in methanol and immunostained with primary antibodies for 1 hr and secondary antibodies for 30 min, both at 37°C. The following primary antibodies were used in the SIM experiments: mouse anti-Ecad antibody (HECD-1, Abcam); rat anti-uvomorulin (DECMA-1, Sigma); mouse anti-Dsg2 antibody (AH12.2, a kind gift from Dr. Asma Nusrat, Emory University); desmoplakin antibody (NW6, a kind gift from Dr. Kathleen Green, Northwestern University). Secondary antibodies conjugated to Alexa Fluorophore were purchased from Invitrogen. SIM was performed using the Nikon N-SIM system on an Eclipse Ti-E microscopy system equipped with a 100x/1.49 NA oil

immersion objective and 488 and 561 nm solid-state lasers. 3D SIM images were captured with an EM charge-coupled device camera (DU-897, Andor Technology) and reconstructed using NIS-Elements software with the N-SIM module (version 3.22, Nikon).

SIM is able to resolve the distance from plaque to plaque when desmosomes are stained with a C-terminal DP antibody and an N-terminal cadherin antibody, as shown in the example SIM image (*Figure 3A*). For analysis of cadherin localization within desmosomes, desmosomes were first defined by regions of parallel DP staining, or 'railroad tracks'. Using ImageJ, a desmosome region of interest (black rectangle) was identified via DP staining (*Figure 3A*). Once a DP and 'railroad track'-positive region of interest was identified, cadherin (red) fluorescence intensity levels were then independently measured along with DP (green) levels. Pairwise multiple comparisons were performed via a Tukey test with a significance level of $\alpha = 0.05$.

## Isolation, culture, transfection and confocal imaging of primary keratinocytes

Spontaneously immortalized primary keratinocytes, isolated from newborn mice, were cultured in DMEM/HAM's F12 (FAD) medium with low $Ca^{2+}$ (50 µM) (Biochrom) supplemented with 10% FCS (chelated), penicillin (100 U ml$^{-1}$), streptomycin (100 µg ml$^{-1}$, Biochrom A2212), adenine ($1.8 \times 10^{-4}$ M, Sigma A3159), L-glutamine (2 mM, Biochrom K0282), hydrocortisone (0.5 µg ml$^{-1}$, Sigma H4001), EGF (10 ng ml$^{-1}$, Sigma E9644), cholera enterotoxin ($10^{-10}$ M, Sigma C-8052), insulin (5 µg ml$^{-1}$, Sigma I1882), and ascorbic acid (0.05 mg ml$^{-1}$, Sigma A4034). Keratinocytes were kept at 32°C and 5% $CO_2$. Ecad$^{KO}$/Pcad$^{KD}$ cells were generated by lentiviral transduction of Ecad-deficient keratinocytes using C14 shRNA directed against Pcad (*Michels et al., 2009*). Cultured cells were regularly monitored for mycoplasma contamination and discarded in case of positive results. Cellular identity was validated by PCR genotyping from genomic DNA and western blot analysis of Dsg 1-2 as markers for keratinocyte identity. In addition, loss of Ecad and efficient knockdown of Pcad was confirmed on the RNA level by RT-PCR as well as on the protein level through western blot analysis.

Ecad-K14E and Ecad-L175D mutants were generated using WT mouse Ecad cDNA in a pcDNA3 backbone including a C-terminal 6myc-tag. Mutations were carried out using 'QuikChange Lightning Site-Directed Mutagenesis Kit' (Agilent). Keratinocytes were transfected at 80–100% confluency with ViromerRed (lipocalyx) according to the manufacturer's protocol. In brief 1.5 µg DNA were diluted in 100 µl Buffer, added to 1.25 µl ViromerRED and incubated for 15 min at room temperature. Approximately 33 µl transfection mix were used per well (24 well plate).

Confocal images were obtained with a Leica TCS SP8, equipped with a white light laser and gateable hybrid detectors (HyDs) and a PlanApo 63x, 1.4 NA objective. Epifluorescence images were obtained with a Leica DMI6000 with a PlanApo 63x, 1.4 NA objective. The following primary antibodies were used in this study: rabbit monoclonal against Dsg2 (1:500, Abcam #ab150372); mouse monoclonal against DP1/2 (1:200, Progen #61003); mouse monoclonal against c-myc (IF 1:2000, Cell Signaling #2276); Secondary antibodies were species-specific antibodies conjugated with either AlexaFluor 488, 594 or 647, used at a dilution of 1:500 for immunofluorescence (Molecular Probes, Life Technologies)

## Acknowledgements

This research was supported in part by the American Heart Association (12SDG9320022) and National Institute of General Medical Sciences of the National Institutes of Health (R01GM121885) to SS; by the Deutsche Forschungsgemeinschaft (DFG SFB 829 A1, DFG SFB 829 Z2, DFG NI 1234/6–1, DFG SPP 1782) to CMN; and by the National Institute of Arthritis and Musculoskeletal and Skin Diseases of the National Institutes of Health (R01AR048266) to APK.

# Additional information

## Funding

| Funder | Grant reference number | Author |
| --- | --- | --- |
| National Institute of Arthritis and Musculoskeletal and Skin Diseases | R01AR048266 | Andrew P Kowalczyk |
| Deutsche Forschungsgemeinschaft | DFG SFB 829 A1 | Carien M Niessen |
| Deutsche Forschungsgemeinschaft | DFG SFB 829 Z2 | Carien M Niessen |
| Deutsche Forschungsgemeinschaft | DFG NI 1234/6-1 | Carien M Niessen |
| Deutsche Forschungsgemeinschaft | DFG SPP 1782 | Carien M Niessen |
| American Heart Association | 12SDG9320022 | Sanjeevi Sivasankar |
| National Institute of General Medical Sciences | R01GM121885 | Sanjeevi Sivasankar |

The funders had no role in study design, data collection and interpretation, or the decision to submit the work for publication.

## Author contributions

Omer Shafraz, Formal analysis, Investigation, Methodology, Writing—original draft; Matthias Rübsam, Sara N Stahley, Formal analysis, Investigation, Methodology, Writing—review and editing; Amber L Caldara, Investigation, Methodology; Andrew P Kowalczyk, Carien M Niessen, Conceptualization, Formal analysis, Supervision, Funding acquisition, Writing—review and editing; Sanjeevi Sivasankar, Conceptualization, Formal analysis, Supervision, Funding acquisition, Writing—original draft, Project administration

## Author ORCIDs

Omer Shafraz (iD) http://orcid.org/0000-0002-2173-1959
Sanjeevi Sivasankar (iD) http://orcid.org/0000-0003-2593-0477

## Decision letter and Author response

Decision letter https://doi.org/10.7554/eLife.37629.012
Author response https://doi.org/10.7554/eLife.37629.013

# Additional files

## Supplementary files

• Transparent reporting form
DOI: https://doi.org/10.7554/eLife.37629.010

## Data availability

All data generated or analyzed during this study are included in the manuscript and supporting files.

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
