## [Decision Letter]

[Editors’ note: a previous version of this study was rejected after peer review, but the authors submitted for reconsideration. The first decision letter after peer review is shown below.]

Thank you for submitting your work entitled "E-cadherin binds to desmoglein and facilitates desmosome assembly" for consideration by *eLife*. Your article has been reviewed by a Senior Editor, a Reviewing Editor, and three reviewers. The reviewers have opted to remain anonymous.

Our decision has been reached after consultation between the reviewers. Based on these discussions and the individual reviews below, we regret to inform you that your work will not be considered further for publication in *eLife*.

This paper addresses the mechanism of cadherin-mediated regulation of desmosome assembly. While the reviewers agree that the work is potentially exciting, they raised several significant issues. First, all agree that the model must be tested by examining co-localization in cells expressing the cadherin mutant. There are also concerns about the reliability of the co-localization data in Figure 5; a more rigorous analysis is needed to demonstrate that the co-localization of E-cadherin and desmoplakin occurs much more than would be expected by chance. Second, the modeling assumes that L175 forms part of the interface, but without assessment of stoichiometry it cannot be ruled out that the loss of binding is an indirect effect of loss of *cis* dimerization. The modeling is also viewed as weak since no complementary interfaces on Dsg2 were tested. Without such data the molecular modeling needs to be considered highly speculative. Finally, as noted by reviewer 3, even if supported by these additions, the model and Discussion section do not really explain how E-cadherin regulates desmosomes.

Given these concerns, the paper cannot be accepted in its current form. However, if these issues can be addressed, we would be happy to consider a resubmission.

Reviewer #1:

In this study Shafraz et al., examine the heterodimerization of E-cadherin and Desmoglein 2 (Dsg2) and its possible role in desmosome formation. They find that E-cadherin can bind to Dsg2 but not Desmocollin, that binding to Dsg2 is blocked by a mutation in E-cadherin that blocks E-cadherin *cis* homodimerization (L175D), but not blocked by an E-cadherin mutation that blocks E-cadherin dimerization in *trans*. These observations, plus computational docking results, support a model in which E-cadherin and Dsg2 interact via a binding interface that is somewhat similar to that used by protocadherins. Fixed-cell SIM immunocytochemistry images of primary keratinocytes are interpreted to indicate that E-cadherin is present in early desmosomes but reduced in mature desmosomes.

Summary recommendation.

The single-molecule biophysical measurements in this paper are for the most part well done, but other data are weak relative to the model that the authors propose. For this reason, the manuscript may not be appropriate for *eLife* in its current form.

Essential revisions:

1) The single-molecule AFM experiments are the most solid part of the paper. However, in my reading the authors do not exclude the possibility that the binding events they observe may be due to interactions between pre-formed E-cadherin *cis*-dimers. The stoichiometry of the interaction is important to establish, since if binding to Dsg2 involved required a E-cadherin *cis*-dimer, it would change the interpretation of the result that the L175D mutation blocks the Dsg2/E-cad interaction. Such a scenario would also challenge the validity of the docking simulations. In my reading, the low probability of binding per contact of the AFM tip with the surface is weak evidence that binding necessarily reflects the interaction of two monomers, nor is the low average density of cadherin molecules on the surface. Further, the cadherins are immobilized with streptavidin, which has multiple biotin binding sites.

For these reasons, I think that proving that heterodimerization occurs between monomers is important. To do so, I would recommend systematically measuring the interaction probability of the tip with the surface as a function of the areal surface density of E-cadherin on the coverslip surface. If the interaction indeed requires monomeric E-cadherin this measurement should scale linearly with added E-cadherin, at least at low densities. However, this specific measurement is not required-any measurement that establishes the stoichiometry will do.

2) The docking simulations are interesting, but in my view offer weak evidence for the specific binding geometry that the authors propose. I would strongly recommend that the authors be more cautious in the interpretation and presentation of these results, or alternatively present additional data, for example EM images, that support or refute the computational result.

3) Unfortunately, the SIM images are not convincing as currently presented. In particular, it is not clear what counts as a "desmosome" in the authors' analysis (Figure 5B). A more sophisticated treatment of colocalization would be required to establish the point that E-cadherin and desmoplakin colocalize, and that this colocalization decreases as junctions mature.

4) I leave this final point at the discretion of the editor. In my view, the colocalization data, even if taken as true, do not establish functional relevance. It would be very nice to see a knockdown/reconstitution experiment with L175D E-cadherin. The prediction is that this molecule would never colocalize with desmoplakin, regardless of the junction age. Violation of this prediction would cast doubt on the authors' favored model.

As a general principle I don't like to add too many additional experiments during the review process. If the editor decides that this or conceptually similar experiments are not required for publication, I would urge the authors to be much more cautious about their interpretive claims.

Reviewer #2:

This is an interesting paper that presents AFM and other data on the interaction of E-cadherin with the desmosomal cadherins Dsg2 and Dsc2, as well as homophilic and heterophilic interaction data for Dsg2 and Dsc2. The authors propose possible mechanisms with relevance to the assembly of desmosomes and suggest models for E-cadherin/Dsg2 interaction, where it is proposed that a novel *trans* (cell-to-cell) interaction between these two proteins may initiate junction formation. Despite high interest in this question, there are issues that diminish my enthusiasm for the work as currently presented, which should be addressed by the authors.

1) The finding that E-cadherin residue L175 is critical for interaction with Dsg2 is intriguing. However, no interacting partner residues are identified. In this circumstance, predicting a conformation for E-cadherin/Dsg2 interaction by docking seems precarious; it seems uncertain that the conformation predicted relates to the true bound conformation.

2) There is only a tenuous connection the proposed Dsg2/E-cadherin interaction to the biology of desmosome assembly. Yet the authors seem to have a clear way forward. They argue that the presence of E-cadherin in nascent desmosomes (assessed experimentally by co-staining for desmoplakin in Figure 5) is evidence for the role of the Dsg2/E-cadherin association they have identified. Now that they have identified a mutant that prevents that association, the experiments of Figure 5 performed with the mutant should lead to different results. This experiment would substantially add to the rigor of this paper.

Reviewer #3:

This manuscript identifies and characterizes a molecular interaction between E-cadherin and Desmoglein 2 using isolated proteins and shows increased colocalization between these two molecules in keratinocytes early during desmosome assembly. These are potentially very interesting observations that may explain how classical cadherins control the assembly of desmosomes. Although this is well documented by several groups in the literature, the underlying mechanisms are still mostly unknown. However, their conclusion "Our integrated single molecule experiments, protein-protein docking predictions and cell based super resolution imaging show that Ecad/Dsg2 interactions play an important role in early desmosome assembly" stated at the start of the Discussion section is a rather strong overstatement given that the interaction is based on characterization on isolated extracellular domains and static albeit high resolution microscopy. No experiment is done to show that when one perturbs this interaction desmosome assembly is indeed impaired, which in my opinion would be the minimal to make this set of data really interesting and relevant for a broad audience, also in light of the fact that the interaction is calcium independent whereas classical cadherin dependent early desmosome assembly does require the presence of calcium (which is not discussed at all).

As this is a fully new interaction of low affinity, it is probably hard to do any interaction assays in the context of cells in which one cell expresses E-cadherin and the other Dsg2 to examine whether there is any interaction. However, as their data are partially based on modeling, the authors should also generate a mutant in which the A125 and A124 are mutated to provide much better evidence for the existence of the proposed of y-dimers.

[Editors’ note: what now follows is the decision letter after the authors submitted for further consideration.]

Thank you for resubmitting your work entitled "E-cadherin binds to desmoglein to facilitate desmosome assembly" for further consideration at *eLife*. Your revised article has been favorably reviewed by Anna Akhmanova (Senior Editor), a Reviewing Editor, and three reviewers.

The manuscript has been improved but there are some remaining issues that need to be addressed before acceptance, as outlined below:

The reviewers feel that while the biophysical data are convincing, more rigorous quantitative analysis is needed to support conclusions made from the data in Figures 3 and 5. In principle these concerns can be addressed with additional data analysis rather than experiments, and in this case, we would consider a revised manuscript.

1) The conclusion that E-cadherin-Dsg interaction is needed for initial desmosome formation is based on colocalization of Dsg and E-cadherin decreasing as the junction matures. However, in Figure 3 it is not clear whether the observed colocalization with DP is significant beyond that expected by chance or arises from time-dependent changes in E-cadherin sub cellular localization unrelated to junction maturation. In Figure 3C, control experiments shown in the right panel highlight co-localization of Dsg2 and DP – known Desmosome components – and show a close correspondence in fluorescence signals at all three time points. In contrast, while the expression of E-cadherin and DP in the left panels does appear to overlap to some degree, they do not show the correspondence in signals observed for the control experiments. The response to the original review points on this subject (e.g. reviewer 1, point 3) focuses on defining a desmosome, but does not address the above point.

2) Similar concerns were raised regarding Figure 5, which is supposed to show that Ecad L175 is essential for efficient intercellular Dsg2 recruitment and desmosome assembly. The images in panel A indeed show a dramatic difference in Dsg2 and DP localization relative to the WT or mutant cadherin. While Dsg2 and DP co-localize with WT or K14E Ecad at junctions, they remain in separate puncta flanking the L175D Ecad junction and do not co-localize with it. However, it is not clear what the authors quantified to test the significance of this phenotype. In B, what is "% of junction forming contacts"? How is a contact defined? What junctions are they measuring? AJ or desmosomes? In C and E, what do they mean by% of Dsg2 or DP positive junctions? How were junctions defined?

There are also modifications to the Introduction and Discussion section needed to address the points above as well as the accessibility of the paper:

1) Related to be points above, the link between in vitro binding experiments and the proposed biological role seems tenuous based on localization studies that reveal overlapped but not truly co-localized staining patterns. The authors need to explain this finding; perhaps co-localization is dynamic, and/or only a subset of E-cadherin molecules is bound to Dsg2. This point at least needs to be discussed to rationalize the results of the localization experiments with the proposed model.

2. In the beginning of the Discussion section the authors claim that they: "identify two critical events that initiate and promote efficient desmosome assembly: (i) stable *trans* homodimerization of Ecad, and (ii) the direct heterophilic binding of Ecad and Dsg2 ectodomains. Our data demonstrates that desmosome assembly is initiated at sites of Ecad *trans* homodimerization. Subsequently, Ecad and Dsg2 bind via a conserved Leu 175 on the Ecad *cis* binding interface and form short-lived heterophilic complexes that localize to early desmosomes and efficiently recruit desmosomal proteins to sites of intercellular contact formation. As desmosomes mature, Dsg2 dissociates from Ecad and forms stable bonds with Dsc2 to mediate robust adhesion." – This paragraph mixes observations, interpretation, and hypothetical models and as such is misleading. They should separate the summary of results from interpretation and their model.

3. The authors write in the Abstract and Introduction that "Ecad interacts with Dsg2 via a conserved Leu 175 on the Ecad *cis* binding interface." – The authors shouldn't assume that readers are familiar with E-cad homophilic interactions and should explain this explicitly.

Reviewer #1:

The core advances of this paper – the demonstration that E-cadherin and Dsg2 bind to each other in AFM experiments, that the interaction is Ca^2+^-independent, and the binding is regulated by the *cis*-interface region of E-cadherin, specifically by residue L175 – are interesting and novel. The model for desmosome assembly based on this finding is interesting, with support from the new SIM experiments (although other conceivable models could also be consistent with these data).

While the new experiments presented in this revised manuscript are squarely aimed at addressing criticisms of biological relevance for the original manuscript, there are issues that remain and should be addressed – at least in the Discussion section. Co-localization data using structured illumination microscopy has been employed to demonstrate E-cadherin and DP co-localization on the cell surface and its changes as desmosomes mature. However, the co-localization is not highly convincing. Specifically, in Figure 3C, control experiments shown in the right panel highlight co-localization of Dsg2 and DP – known Desmosome components – and show a close correspondence in fluorescence signals at all three time points. In contrast, while the expression of E-cadherin and DP in the left panels does appear to overlap to some degree, they do not show the correspondence in signals observed for the control experiments (compare traces in Figure 3C). Changes in E-cadherin/DP localization upon desmosome maturation described in the text are also not clearly evident in the Figure 3.

In Figure 5, the data show a dependence of E-cadherin and Dsg2 on DP localization in transfected keratinocytes. X-dimer and *cis*-interface mutant E-cadherins were tested in this assay and the *cis*-mutant appears to interfere with co-localization of E-cadherin and Dsg2 or DP. Again, though, co-localization of wild-type E-cadherin with Dsg2 does not seem convincing (Figure 5A top panels). The change caused by the L175D mutation is striking (Figure 5A bottom panel), but does not necessarily implicate binding to Dsg2, as this mutation is expected to disrupt E-cadherin *cis* interactions, which could also have phenotypic effects.

Figure 2 could easily be supplementary, since it is hard for a non-specialist to glean any information from the plots. The numbers quoted in the text are sufficient. In addition, the authors mention that the k_off_ rate is 10 fold weaker for the Dsc2/Dsg2 dimer, yet the values quoted in the text only suggest a 6-fold difference. This should be clarified.

Overall, this paper presents an interesting biophysical study of E-cadherin and Dsg2 binding, the link between in vitro binding experiments and the proposed biological role seems tenuous based on localization studies that reveal overlapped but not truly co-localized staining patterns. The authors need to explain this finding; perhaps co-localization is dynamic, and/or only a subset of E-cadherin molecules is bound to Dsg2. This point at least needs to be discussed to rationalize the results of the localization experiments with the proposed model.

Reviewer #2:

The authors have addressed many of my prior concerns. My remaining concern is that they do not show that the localization of E-cadherin to DP puncta is more (or less) than what would be expected by chance. The trend with time that they report is not sufficient, as in principle the local abundance of E-cadherin could be changing in time.

Reviewer #3:

In their paper, "E-cadherin binds to desmoglein to facilitate desmosome assembly", Shafraz et al., use single molecule AFM to show that E-cadherin forms a direct, calcium independent, interaction with Dsg2 that has a lifetime similar to Dsc2/Dsc2 dimers but is ten times weaker than Dsg2/Dsc2 dimers. On the E-cadherin side, this interaction depended on Leu175, which also mediates Ecad *cis* interactions, but was independent of the residues that mediate *trans* Ecad interactions. The authors used SIM imaging of human keratinocytes, following a switch from low to high calcium, to show that Ecad levels in Desmoplakin positive structures are higher in early desmosome than in mature desmosomes. Finally, using a model of EcadKO/PcadKD mouse keratinocytes, in which they reintroduce WT or mutant Ecad, the authors show using conventional microscopy that Ecad(L175D) is highly ineffective in recruiting Dsg2 at early time points and in desmosome formation, and this defect is not explained by its compromised ability to form AJs.

Overall, this work is novel and interesting and the combination of biophysical and cell biological approaches is very powerful and provides molecular mechanistic insight into age-old observations of interaction between classical and desmosomal cadherins. After addressing the points outlined below, I think this work will be most appropriate for publication in *eLife*.

1) The most burning question I was left with after reading the paper, and which I think the authors can and should answer before publication is whether the interaction between Ecad and Dsg2 is a *cis* or *trans* interaction. While reading the paper, I had imagined a *trans* interaction and it was only when I reached Figure 6 and saw their model that I realized the authors assume it to be a *cis* interaction, though they say in the discussion their data is equally consistent with a *trans* interaction. I feel this is an important point that can't be left open. Two color STORM imaging or biochemical cross-linking experiments could be employed to tease apart these two possibilities.

2) Figure 3 – If E-cad is being replaced by Dsc2, then Dsc2 level should go up over time. Can the authors show this?

3) Given the focus of this paper is on the interaction between Ecad and Dsg2, it seems to me to be important to image both endogenous E-cad and Dsg2 with SIM and show their transient co-localization in desmosomes.

4) Figure 5 is supposed to show that Ecad L175 is essential for efficient intercellular Dsg2 recruitment and desmosome assembly. The images in panel A indeed show a dramatic difference in Dsg2 and DP localization relative to the WT or mutant cadherin. While Dsg2 and DP co-localize with WT or K14E Ecad at junctions, they remain in separate puncta flanking the L175D Ecad junction and do not co-localize with it. However, it is not clear what the authors quantified to test the significance of this phenotype. In B, what is "% of junction forming contacts"? How is a contact defined? What junctions are they measuring? AJ or desmosomes? In C and E, what do they mean by% of Dsg2 or DP positive junctions? How were junctions defined?

Neither of these quantifications seems to take advantage of the rich 2D (or 3D) information in the images to quantify co-localization.

5) In the beginning of the Discussion section the authors claim that they: "identify two critical events that initiate and promote efficient desmosome assembly: (i) stable *trans* homodimerization of Ecad, and (ii) the direct heterophilic binding of Ecad and Dsg2 ectodomains. Our data demonstrates that desmosome assembly is initiated at sites of Ecad *trans* homodimerization. Subsequently, Ecad and Dsg2 bind via a conserved Leu 175 on the Ecad *cis* binding interface and form short-lived heterophilic complexes that localize to early desmosomes and efficiently recruit desmosomal proteins to sites of intercellular contact formation. As desmosomes mature, Dsg2 dissociates from Ecad and forms stable bonds with Dsc2 to mediate robust adhesion." – This paragraph mixes observations, interpretation, and hypothetical models and as such is misleading. They should separate the summary of results from interpretation and their model.

---

## [Author Response]

[Editors’ note: the author responses to the first round of peer review follow.]

1) This paper addresses the mechanism of cadherin-mediated regulation of desmosome assembly. While the reviewers agree that the work is potentially exciting, they raised several significant issues. First, all agree that the model must be tested by examining co-localization in cells expressing the cadherin mutant.

We thank the reviewers for suggesting these important experiments. As described in our detailed response to the reviewers, we have now tested the role of the different Ecad mutations in impeding recruitment of Dsg2 in keratinocytes. We expressed full-length Ecad WT, and Ecad mutants (L175D, K14E, and W2A-K14E) in Ecad-knockout, Pcad-knockdown mouse keratinocytes (E^KO^/P^KD^) and analyzed Dsg2 recruitment to sites of intercellular contacts. These results confirm that amino acid L175 on Ecad, mediates Dsg2 interactions and facilitates early desmosome complex formation in cells. Our experiments also reveal that desmosome assembly is initiated at sites of Ecad *trans* homodimerization.

2) There are also concerns about the reliability of the co-localization data in Figure 5; a more rigorous analysis is needed to demonstrate that the co-localization of E-cadherin and desmoplakin occurs much more than would be expected by chance.

In the field, desmosomes are identified by the size and rail road track appearance of DP staining in SIM images. As outlined in our response to the reviewers, we used this established approach to locate desmosomes. In addition, we now stress that we are not measuring colocalization but rather the recruitment of different cadherins (Ecad and Dsg) to regions of the membrane that display this pattern.

3) Second, the modeling assumes that L175 forms part of the interface, but without assessment of stoichiometry it cannot be ruled out that the loss of binding is an indirect effect of loss of cis dimerization. The modeling is also viewed as weak since no complementary interfaces on Dsg2 were tested. Without such data the molecular modeling needs to be considered highly speculative.

As described in our detailed response to the reviewers, previous computational simulations and biophysical experiments show that Ecad *cis*-dimer formation requires prior *trans* dimerization. Consequently, Ecads cannot bind to the surface as stand-alone, pre-formed, *cis-*dimers. This makes it exceedingly unlikely that the measured binding interactions occur between Ecad *cis*-dimers and Dsg2. Nonetheless, we agree with the reviewers that the molecular docking results are quite weak. We have therefore eliminated these modeling results from the manuscript.

4) Finally, as noted by reviewer 3, even if supported by these additions, the model and Discussion section do not really explain how E-cadherin regulates desmosomes.

Based on our new cellular structure-function data, we now propose a model whereby desmosome assembly is facilitated both by the direct *cis* interactions of Ecad and Dsg2 and the *trans* binding of opposing Ecad. In our model, the *trans* homodimerization of Ecad from opposing cells serves as a spatial cue to coordinate desmosome assembly. Ecad subsequently forms *cis*-dimer complexes with Dsg2 to initiate desmosome assembly. The model is presented and discussed in the new manuscript.

Reviewer #1:1) The single-molecule AFM experiments are the most solid part of the paper. However, in my reading the authors do not exclude the possibility that the binding events they observe may be due to interactions between pre-formed E-cadherin cis-dimers. The stoichiometry of the interaction is important to establish, since if binding to Dsg2 involved required a E-cadherin cis-dimer, it would change the interpretation of the result that the L175D mutation blocks the Dsg2/E-cad interaction. Such a scenario would also challenge the validity of the docking simulations. In my reading, the low probability of binding per contact of the AFM tip with the surface is weak evidence that binding necessarily reflects the interaction of two monomers, nor is the low average density of cadherin molecules on the surface. Further, the cadherins are immobilized with streptavidin, which has multiple biotin binding sites.For these reasons, I think that proving that heterodimerization occurs between monomers is important. To do so, I would recommend systematically measuring the interaction probability of the tip with the surface as a function of the areal surface density of E-cadherin on the coverslip surface. If the interaction indeed requires monomeric E-cadherin this measurement should scale linearly with added E-cadherin, at least at low densities. However, this specific measurement is not required-any measurement that establishes the stoichiometry will do.

Previous computer simulations (Wu et al., 2010; Wu et al., 2011) have shown that Ecad *cis* homodimerization requires prior Ecad *trans*-dimer formation. Similarly, previous single molecule FRET measurements have shown that it is not possible to form stand-alone Ecad *cis*-dimers, even when Ecad monomers are placed in close proximity in a *cis* orientation (Zhang et al., 2009). Taken together, these results suggest that in our experiments, Ecads cannot be immobilized on the AFM tip or substrate as pre-formed *cis*-dimers. Consequently, the failure of the Ecad-L175D mutant to interact with Dsg2 is not due to the absence of Ecad *cis* dimerization. We now discuss this in subsection “Leu 175 mediates Ecad and Dsg2 interactions” of the manuscript.

Unfortunately, technical limitations do not allow the protein stoichiometry experiment that the reviewer suggests. In our experiments, we use protein surface densities that allow us to unambiguously identify single binding events from the corresponding stretching of PEG tethers. If the cadherin surface density is increased, as proposed by the reviewer, the probability of simultaneous multiple Ecad binding events increases which makes a quantitative analysis of protein binding probability unreliable. For this reason (along with the intrinsic weakness of our docking analysis), we have excluded the protein docking simulations from the manuscript.

Finally, in the revised manuscript, we now use cluster analysis to group the single molecule unbinding events for Dynamic Force Spectroscopy (DFS). We have recently shown that the K-means clustering algorithm that we employed, greatly improves the estimation of kinetic parameters in DFS (Yen and Sivasankar, 2018). Consequently, the lifetimes we now report for Ecad/Dsg2, Dsc2/Dsg2, and Dsc2/Dsc2 interactions are more reliable.

2) The docking simulations are interesting, but in my view offer weak evidence for the specific binding geometry that the authors propose. I would strongly recommend that the authors be more cautious in the interpretation and presentation of these results, or alternatively present additional data, for example EM images, that support or refute the computational result.

We agree with the reviewer that the protein docking data is weak. We have therefore eliminated these results from the manuscript.

3) Unfortunately, the SIM images are not convincing as currently presented. In particular, it is not clear what counts as a "desmosome" in the authors' analysis (Figure 5B). A more sophisticated treatment of colocalization would be required to establish the point that E-cadherin and desmoplakin colocalize, and that this colocalization decreases as junctions mature.

In previous studies (Stahley et al., 2016a, 2016b) we have utilized the “rail road track” appearance of DP as revealed by super-resolution imaging to define desmosomes by immunofluorescence (as depicted in Figure 3A). Others in the field have also used ‘railroad’ track morphology to identify desmosomes in SIM images (see for instance: Chen et al., (2012); Ungewiß et al., (2017)). The size and organization of these structures, as revealed by SIM, is fully consistent with classical EM definitions of desmosomes. We measured pixel intensity for the two cadherins (Ecad or Dsg2) within these structures using standard image analysis. Importantly, we are not attempting to claim co-localization. Rather, we are measuring recruitment of cadherin to membrane domains that display DP rail road track staining patterns. We do appreciate that it is sometimes difficult to discern rail-road tracks at early time points. For this reason, the earliest time we felt that we could identify bona fine rail road track staining was at 1 hour. Furthermore, we are only making measurements at locations where we can observe rail road track patterns, rather than DP puncta. Lastly, our finding that Ecad is present in nascent desmosomes is consistent with classic immuno-EM experiments showing that Ecad can localize to desmosomes (Jones, 1988). Therefore, we feel confident in our ability to identify desmosomes and to measure relative levels of the two cadherins at different times.

4) I leave this final point at the discretion of the editor. In my view, the colocalization data, even if taken as true, do not establish functional relevance. It would be very nice to see a knockdown/reconstitution experiment with L175D E-cadherin. The prediction is that this molecule would never colocalize with desmoplakin, regardless of the junction age. Violation of this prediction would cast doubt on the authors' favored model.

Based on the reviewer’s suggestion, we have now performed experiments to directly test the role of the different Ecad extracellular domain mutations in impeding recruitment of DP and Dsg2 in keratinocytes. This data is described in subsection “Ecad L175 is essential for efficient intercellular Dsg2 recruitment and desmosome assembly”. The corresponding images are shown in Figure 5 and in Figure 5—figure supplement 1. The methods used are described in subsection “Isolation, culture, transfection and confocal imaging of primary keratinocytes”. For these experiments, we used E^KO^/P^KD^ mouse keratinocytes, which express virtually no classical cadherins. We have previously shown that these E^KO^/P^KD^ keratinocytes are unable to assemble AJs and desmosomes due to the loss of all classical cadherins (Michels et al., 2009). These cells thus allow us to directly assess the ability of the Ecad mutants to *i)* initiate AJ formation and *ii)* assess their ability to recruit desmosomal components to sites of initial cell-cell contact formation.

Since we were interested in assessing the ability of Ecad mutants to recruit desmosomal proteins early during junction formation, we expressed either full-length Ecad WT or mutants in E^KO^/P^KD^ keratinocytes and analyzed DP and Dsg2 recruitment to sites of intercellular contacts using confocal microscopy. To examine recruitment at different stages of junction formation and maturation, keratinocytes were fixed and immunostained for DP, Dsg and Ecad at three time points following the Ca^2+^ switch (3 hours, 6 hours and 18 hours).

Our data showed that 3 hours after the Ca^2+^ switch, 93% of WT-Ecad was enriched in zipper-like early AJs at sites of intercellular contacts. In contrast only 48% of L175D-Ecad transfected keratinocytes formed AJ zippers, likely due to impaired Ecad *cis*-dimer formation (Figure 5 A, B). At these early time points following the Ca^2+^ switch, 66% and 91% of the WT-Ecad zipper contacts were positive for Dsg2 and DP respectively (Figure 5 A, C, D, E). When the cells were allowed to engage in Ca^2+^-dependent intercellular adhesion for 18 hour, the junctional localizations of Ecad-Dsg2 and Ecad-DP increased to 97% and 100% respectively (Figure 5 A, C, D, E). In contrast, only 39% of L175-induced zipper contacts showed Dsg2 recruitment 3 hours after switching to high Ca^2+^ which increased to 65% at 18 hours (Figure 5 A, C), confirming that a direct interaction between Ecad and Dsg2 is required for efficient recruitment of Dsg2 to early intercellular contacts. Similarly, after 3 hours, only 28% of L175D-Ecad mutant established intercellular contacts were positive for DP, which was increased to 72% after 18 hours, suggesting a compensatory mechanism at later stages (Figure 5 D, E).

To confirm that the observed effect of the L175D mutation was not due to hindered AJ formation, we transfected the E^KO^/P^KD^ keratinocytes with the full-length Ecad-K14E mutant that abolishes X-dimer formation and traps Ecad in a strand-swap dimer conformation. Our data showed that only 4% of the transfected contacting cells formed zippers at early (3 hours) timepoints after switching to high Ca^2+^ with only 24% of the transfected contacting cells showing zippers at late (18 hours) time points (Figure 5 A, B, Figure 5—figure supplement 1), confirming that K14 is essential for effective AJ formation, and thus intercellular contact establishment. However, the few K14E AJs that were formed, recruited Dsg2, and especially DP, more efficiently than L175D (Figure 5 A, C, E, Figure 5—figure supplement 1). This result confirmed that delayed recruitment of Dsg2 to inter-cellular contacts was not the result of the inability of the Ecad L175D to efficiently form AJs but rather due to absence of direct interaction between Ecad and Dsg2. As there was never any co-localization of DP and Ecad in the absence of zippers (Figure 5—figure supplement 1), the data also suggest that Ecad *trans* interactions precede Ecad/Dsg2 interactions. This conclusion is further strengthened by our finding that no DP recruitment was observed upon abolishing Ecad *trans* adhesion, and thus zippers, by transfecting the full-length Ecad-DM (W2A-K14E double mutant, Figure 5—figure supplement 1) in E^KO^/P^KD^ keratinocytes. Taken together these results confirm that amino acid L175 mediates Dsg2 interactions and facilitates early desmosome complex formation in cells.

Reviewer #2:1) The finding that E-cadherin residue L175 is critical for interaction with Dsg2 is intriguing. However, no interacting partner residues are identified. In this circumstance, predicting a conformation for E-cadherin/Dsg2 interaction by docking seems precarious; it seems uncertain that the conformation predicted relates to the true bound conformation.

We agree with the reviewer that the protein docking data is speculative and have therefore eliminated these results from the manuscript.

2) There is only a tenuous connection the proposed Dsg2/E-cadherin interaction to the biology of desmosome assembly. Yet the authors seem to have a clear way forward. They argue that the presence of E-cadherin in nascent desmosomes (assessed experimentally by co-staining for desmoplakin in Figure 5) is evidence for the role of the Dsg2/E-cadherin association they have identified. Now that they have identified a mutant that prevents that association, the experiments of Figure 5 performed with the mutant should lead to different results. This experiment would substantially add to the rigor of this paper.

As described in response 4 to reviewer 1, we expressed full-length Ecad WT, and Ecad mutants in E^KO^/P^KD^ keratinocytes and analyzed Dsg2 and DP recruitment to sites of intercellular contacts. These results confirm that amino acid L175 mediates Dsg2 interactions and facilitates early desmosome complex formation in cells. This data is described in subsection “Ecad L175 is essential for efficient intercellular Dsg2 recruitment and desmosome assembly” of the manuscript. The corresponding images are shown in Figure 5 and in Figure 5—figure supplement 1. The methods used are described in subsection “Isolation, culture, transfection and confocal imaging of primary keratinocytes”.

Reviewer #3:1) This manuscript identifies and characterizes a molecular interaction between E-cadherin and Desmoglein 2 using isolated proteins and shows increased colocalization between these two molecules in keratinocytes early during desmosome assembly. These are potentially very interesting observations that may explain how classical cadherins control the assembly of desmosomes. Although this is well documented by several groups in the literature, the underlying mechanisms are still mostly unknown. However, their conclusion "Our integrated single molecule experiments, protein-protein docking predictions and cell based super resolution imaging show that Ecad/Dsg2 interactions play an important role in early desmosome assembly" stated at the start of the Discussion section is a rather strong overstatement given that the interaction is based on characterization on isolated extracellular domains and static albeit high resolution microscopy. No experiment is done to show that when one perturbs this interaction desmosome assembly is indeed impaired, which in my opinion would be the minimal to make this set of data really interesting and relevant for a broad audience.

As described in response 4 to reviewer 1, we have now expressed full-length Ecad WT, and Ecad mutants in E^KO^/P^KD^ keratinocytes and shown that amino acid L175 mediates Dsg2 interactions and facilitates early desmosome formation in cells as assessed by DP recruitment. This data is described in subsection “Ecad L175 is essential for efficient intercellular Dsg2 recruitment and desmosome assembly” of the manuscript. The corresponding images are shown in Figure 5 and in Figure 5—figure supplement 1. The methods used are described in subsection “Isolation, culture, transfection and confocal imaging of primary keratinocytes”.

2) Also, in light of the fact that the interaction is calcium independent whereas classical cadherin dependent early desmosome assembly does require the presence of calcium (which is not discussed at all).

We thank the reviewer for this insightful comment. As described in subsection “Ecad L175 is essential for efficient intercellular Dsg2 recruitment and desmosome assembly”, confocal imaging of mutant Ecad expressed in mouse keratinocytes now reveals that desmosome assembly is initiated at sites of Ca^2+^ dependent Ecad *trans* homodimerization. Based on our data, we propose a model whereby desmosome assembly is facilitated both by the direct *cis* interactions of Ecad and Dsg2 and the *trans* binding of opposing Ecad. In our model, the *trans* homodimerization of Ecad from opposing cells serves as a spatial cue to coordinate desmosome assembly. Ecad subsequently forms *cis* dimer complexes with Dsg2. Once localized within the native desmosome, Dsg2 dissociates from Ecad and binds to Dsc2 to form mature desmosomes. Since the Dsc2/Dsg2 complex has a longer lifetime than both the Ecad/Dsg2 complex and Dsc2/Dsc2 complex, this likely allows for robust cell-cell adhesion and permits the mature desmosome to withstand mechanical force. This model is described in Figure 6, and in the Discussion section.

3) As this is a fully new interaction of low affinity, it is probably hard to do any interaction assays in the context of cells in which one cell expresses E-cadherin and the other Dsg2 to examine whether there is any interaction. However, as their data are partially based on modeling, the authors should also generate a mutant in which the A125 and A124 are mutated to provide much better evidence for the existence of the proposed of y-dimers.

Since the protein docking results are not robust, we have now eliminated these data from the manuscript. However, our experiments with E^KO^/P^KD^ keratinocytes showing that mutation of amino acid L175 on Ecad impairs recruitment of Dgs2 and especially DP to sites of early contact formation, demonstrates that Ecad-L175 facilitates desmosome formation in cells, confirms our biophysical results.

[Editors' note: the author responses to the re-review follow.]

Summary:The reviewers feel that while the biophysical data are convincing, more rigorous quantitative analysis is needed to support conclusions made from the data in Figures 3 and 5. In principle these concerns can be addressed with additional data analysis rather than experiments, and in this case, we would consider a revised manuscript.

We thank the editor for this encouraging review. As described below, our revised manuscript addresses all the highlighted issues.

1) The conclusion that E-cadherin-Dsg interaction is needed for initial desmosome formation is based on colocalization of Dsg and E-cadherin decreasing as the junction matures. However, in Figure 3 it is not clear whether the observed colocalization with DP is significant beyond that expected by chance or arises from time-dependent changes in E-cadherin sub cellular localization unrelated to junction maturation. In Figure 3C, control experiments shown in the right panel highlight co-localization of Dsg2 and DP – known Desmosome components – and show a close correspondence in fluorescence signals at all three time points. In contrast, while the expression of E-cadherin and DP in the left panels does appear to overlap to some degree, they do not show the correspondence in signals observed for the control experiments. The response to the original review points on this subject (e.g. reviewer 1, point 3) focuses on defining a desmosome, but does not address the above point.

We would like to clarify that we are not measuring co-localization but instead the recruitment of either Ecad or Dsg2 to regions of the membrane that display a DP ‘railroad track’ pattern. To make this clearer, we have now modified the revised manuscript to emphasize that Ecad is ‘enriched in nascent desmosomes’ rather than being ‘excluded from mature desmosomes’.

To directly address the reviewers concerns regarding our measurement of Ecad enrichment in desmosomes, we now show that the relative levels of Ecad along the entire cell border (Ecad:DP ratio at cell borders) remain unchanged over time. This data confirms that the Ecad enrichment within DP railroad tracks at early time points is specific to desmosomal regions of the membrane and is significant beyond that expected by random chance or because of changes in E-cadherin localization unrelated to junction maturation. The data is shown in Figure 3—figure supplement 1 and is described in subsection “Ecad is present in nascent desmosomes but not in mature desmosomes”.

Finally, since the line-scans shown in Figure 3 were confusing, we have eliminated the line-scans from the revised manuscript. Instead, in Figure 3B, we now show several representative images of desmosomal regions in human keratinocytes cultured in high Ca^2+^ media for 1, 3 or 18 hours. These results emphasize the main ‘take-away’ message of Figure 3C: Ecad levels are enriched in nascent desmosomes, with relative levels decreasing as desmosomes mature.

2) Similar concerns were raised regarding Figure 5, which is supposed to show that Ecad L175 is essential for efficient intercellular Dsg2 recruitment and desmosome assembly. The images in panel A indeed show a dramatic difference in Dsg2 and DP localization relative to the WT or mutant cadherin. While Dsg2 and DP co-localize with WT or K14E Ecad at junctions, they remain in separate puncta flanking the L175D Ecad junction and do not co-localize with it. However, it is not clear what the authors quantified to test the significance of this phenotype. In B, what is "% of junction forming contacts"? How is a contact defined? What junctions are they measuring? AJ or desmosomes? In C and E, what do they mean by% of Dsg2 or DP positive junctions? How were junctions defined?

We have now added a figure panel (Figure 5—figure supplement 1A) to illustrate the quantification criteria and junctional categories that were used in our analysis. This panel shows examples of cells transfected with Ecad mutant that do not (left panel) or do (right panel) show intercellular AJ formation. For Dsg2 or DP recruitment, we only quantified those intercellular interfaces in which AJ zippers were observed. In the insets of the figure panel, we show examples of AJ positive contacts that are either positive for DP or negative for DP.

In subsection “Ecad L175 is essential for efficient intercellular Dsg2 recruitment and desmosome assembly” of the manuscript, we now state that the ability of transfected keratinocytes to form AJs was first assessed by the formation of ‘zipper-like’ patterns of Ecad at intercellular contacts. We have previously shown that these zippers represent early AJs and recruit AJ marker proteins like vinculin (Rübsam et al., 2017). We only examined intercellular interfaces with AJ zippers, for their ability to recruit Dsg2 and DP to these contacts. Importantly, we have never observed any enrichment of desmosomal components at intercellular contacts in the absence of AJ zippers (Michels et al., 2009).

As described in response 3a below, we now discuss in the Discussion section that although Dsg2 and DP do show some co-localization at Ecad-positive zippers within 3 hours, there is also non-overlapping junctional staining suggesting that AJs and desmosomes are already starting to segregate into distinct intercellular junctions. We did not quantify this co-localization in our manuscript, since the main point of our experiments were to show the relevance of L175 for recruiting Dsg2 to intercellular interfaces and facilitating desmosome formation (as illustrated by DP recruitment to intercellular contacts). We believe that the delay in recruitment between WT, L175 mutant, and the K14 mutant, that is quantified in Figure 5, illustrates this point.

3) There are also modifications to the Introduction and Discussion section needed to address the points above as well as the accessibility of the paper:

*a) Related to be points above, the link between* in vitro *binding experiments and the proposed biological role seems tenuous based on localization studies that reveal overlapped but not truly co-localized staining patterns. The authors need to explain this finding; perhaps co-localization is dynamic, and/or only a subset of E-cadherin molecules is bound to Dsg2. This point at least needs to be discussed to rationalize the results of the localization experiments with the proposed model.*

Due to the transient nature of Ecad/Dsg2 interactions, desmosomal proteins recruited to junctions are expected to rapidly segregate from Ecad. Consequently, images of intercellular interfaces are expected to have both overlapping and separate Ecad and desmosomal protein staining patterns. In agreement, although Dsg2 and DP show some co-localization at Ecad-positive zippers (Figure 5), significant non-overlapping junctional staining is observed, even within the first 3 hours. This suggests that AJs and desmosomes rapidly segregate into distinct intercellular junctions. We now state this in the Discussion section.

b) In the beginning of the Discussion section the authors claim that they: "identify two critical events that initiate and promote efficient desmosome assembly: (i) stable trans homodimerization of Ecad, and (ii) the direct heterophilic binding of Ecad and Dsg2 ectodomains. Our data demonstrates that desmosome assembly is initiated at sites of Ecad trans homodimerization. Subsequently, Ecad and Dsg2 bind via a conserved Leu 175 on the Ecad cis binding interface and form short-lived heterophilic complexes that localize to early desmosomes and efficiently recruit desmosomal proteins to sites of intercellular contact formation. As desmosomes mature, Dsg2 dissociates from Ecad and forms stable bonds with Dsc2 to mediate robust adhesion." – This paragraph mixes observations, interpretation, and hypothetical models and as such is misleading. They should separate the summary of results from interpretation and their model.

As suggested by the reviewers, we have eliminated the interpretations from the first paragraph of the Discussion section and now only summarize the results.

c) The authors write in the Abstract and Introduction "Ecad interacts with Dsg2 via a conserved Leu 175 on the Ecad cis binding interface." – The authors shouldn't assume that readers are familiar with E-cad homophilic interactions and should explain this explicitly.

In the Abstract, we now state that ‘Previous studies demonstrate that E-cadherin (Ecad), an adhesive protein that interacts in both *trans* and *cis* conformations, facilitates desmosome assembly via an unknown mechanism’.

Furthermore, in the Introduction, we now state: ‘Since Ecad interacts laterally to form *cis-*dimers on the same cell surface (Harrison et al., 2011) while Ecad molecules from opposing cells interact in a *trans* strand-swap dimer conformation (Boggon et al., 2002; Parisini et al., 2007; Vendome et al., 2011) and a *trans* X-dimer conformation (Ciatto et al., 2010; Harrison et al., 2010), we used mutants that specifically abolish either Ecad *trans* or *cis* interactions and tested their binding to either Dsg2 or Dsc2’. Finally, in the introduction, we now state that ‘Previous structural studies have shown that L175 mediates homophilic Ecad *cis* dimerization (Harrison et al., 2011)’.